# Communication Bounds for the Distributed Experts Problem

**Zhihao Jia**
Carnegie Mellon University
zhihao@cmu.edu

**Qi Pang**
Carnegie Mellon University
qipang@cmu.edu

**Trung Tran**
University of Pittsburgh
tbt8@pitt.edu

**David Woodruff**
Carnegie Mellon University
dwoodruf@cs.cmu.edu

**Zhihao Zhang**
Carnegie Mellon University
zhihaoz3@cs.cmu.edu

**Wenting Zheng**
Carnegie Mellon University
wenting@cmu.edu

## Abstract

In this work, we study the experts problem in the distributed setting where an expert's cost needs to be aggregated across multiple servers. Our study considers various communication models such as the message-passing model and the broadcast model, along with multiple aggregation functions, such as summing and taking the $\ell_p$ norm of an expert's cost across servers. We propose the first communication-efficient protocols that achieve near-optimal regret in these settings, even against a strong adversary who can choose the inputs adaptively. Additionally, we give a conditional lower bound showing that the communication of our protocols is nearly optimal. Finally, we implement our protocols and demonstrate empirical savings on the HPO-B benchmarks.

## 1 Introduction

Online prediction with expert advice is an indispensable task in many fields, including bandit learning (Auer et al., 2002; Lattimore & Szepesvári, 2020), online optimization (Shalev-Shwartz et al., 2012; Hazan et al., 2016), robot control (Doyle et al., 2013), and financial decision making (Dixon et al., 2020). The problem involves $n$ experts making individual predictions and receiving corresponding costs on each of $T$ days. On each day, we choose an expert based on the historical costs of the experts on previous days, and we receive the cost of the selected expert on that day. The objective is to compete with the best single expert in hindsight, i.e., to minimize the average *regret*, defined as the additional cost the algorithm incurs against the best expert in a horizon of $T$ days. It is known that the *Exponential Weights Algorithm* (EWA) and *Multiplicative Weight Update* (MWU) method achieve an optimal regret of $O(\sqrt{\frac{\log n}{T}})$ given all historical information, even in the presence of a strong adversary Arora et al. (2012). With less information, the *exponential-weight algorithm for exploration and exploitation* (Exp3) achieves near-optimal regret $O(\sqrt{\frac{n \log n}{T}})$ in the adversarial bandit setup, where only the cost of one expert is observed on a single day.

For a large number of experts and days, it may not be feasible to run classical low-regret algorithms. Motivated by this, recent work (Srinivas et al., 2022; Peng & Zhang, 2022; Woodruff et al., 2023; Peng & Rubinstein, 2023; Aamand et al., 2023) considers the experts problem in the *data stream model*, where the expert predictions are typically streamed through main memory, and a small summary of historical information is stored.

In this paper, we consider an alternative model in the big data setting, namely, the distributed model, where expert costs are split across $s$ servers, and there is a central coordinator who can run a low-regret

Table 1: Summary of our constant probability communication upper bounds.

| | Upper bounds w/ a constant probability | | | |
|---|---|---|---|---|
| **Algorithms** | DEWA-S | DEWA-M | DEWA-L | DEWA-L |
| **Agg Func** | SUM | MAX | $\ell_{p>2}$ | $\ell_p(1+\epsilon < p \le 2)$ |
| **Broadcast** | $\tilde{O}(\frac{n}{R^2}) + O(Ts)$ | $\tilde{O}(\frac{n}{R^2}+Ts)$ | $\tilde{O}(\frac{n}{R^2}+Ts)$ | $\tilde{O}(\frac{n}{R^{1+1/\epsilon}}+Ts)$ |
| **Message-Passing** | | - | - | - |

Table 2: Summary of our high probability communication upper bounds.

| | Upper bounds w/ probability $1 - 1/\text{poly}(T)$ | | | |
|---|---|---|---|---|
| **Algorithms** | DEWA-S-P | DEWA-M-P | DEWA-L-P | DEWA-L-P |
| **Agg Func** | SUM | MAX | $\ell_{p>2}$ | $\ell_p(1+\epsilon < p \le 2)$ |
| **Broadcast** | $\tilde{O}(\frac{n}{R^2}+Ts)$ | $\tilde{O}(\frac{n}{R^2}+Ts)$ | $\tilde{O}(\frac{n}{R^2}+Ts)$ | $\tilde{O}(\frac{n}{R^{1+1/\epsilon}}+Ts)$ |
| **Message-Passing** | | - | - | - |

algorithm. However, communicating with different servers is expensive, and the goal is to design a low communication protocol that achieves low regret.

A *motivating example* is a distributed online optimization problem, where different servers hold different samples, and each expert could correspond to a different model in an optimization problem over the union of the samples as in the HPO-B real-world benchmark (Arango et al., 2021). In this case, it is natural for the cost of an expert to be the sum of the costs of the expert across all servers. The goal is thus to minimize the cumulative costs in an online fashion by choosing models on a daily basis. Another example of an aggregation function could be the maximum across servers; indeed, this could be useful if there is a maximum tolerable cost on the servers, which we would like not to exceed. For our lower bounds, we also ask the protocol to be able to tell at least if the cost of the expert it chose on a given day is non-zero; this is a minimal requirement of all existing algorithms, such as MWU or Exp3, which update their data structure based on such a cost. It is also desirable in applications such as the experts problem where one wants to know if the prediction was right or wrong.

In our setting, a coordinator needs to choose an expert based on historical interactions with $s$ servers each day. We focus on two widely studied communication models, namely, the *message-passing model* with two-way communication channels and the *broadcast model* with a broadcast channel. In the message-passing model, the coordinator initiates a round of interaction with a given server, and the messages exchanged are only seen by the coordinator and that particular server. The coordinator then decides who speaks next and repeats this process. The broadcast model is also commonly studied in practice and theory. It can be viewed as a model for single-hop wireless networks. In the broadcast model, each message exchanged is seen by all servers and the coordinator. We note that the broadcast model was a central communication model studied for clustering in Chen et al. (2016).

As in the distributed online learning setup, we can view each server as a database, where it possibly receives new data daily. The costs of the $n$ experts on a day then correspond to $n$ possibly different functions of the data on that day. We note that the costs may be explicitly given or implicit functions of the data, and if the latter, they may only need to be computed as required by the protocol.

We aim to achieve a near-optimal regret versus communication tradeoff in this setting over a horizon of $T$ days. Given the memory-efficient streaming algorithms of Srinivas et al. (2022); Peng & Zhang (2022) and the close connection between streaming algorithms and communication-efficient protocols, one might think that implementing a streaming algorithm in our settings is optimal. While we could run a streaming algorithm, a critical difference here is that the coordinator is not memory-bounded and thus can afford to store a weight for each expert. While it cannot run EWA or MWU, which would require $\Omega(sn)$ communication per day, it can run a distributed Exp3 algorithm, which samples a single expert and thus has low communication, but maintains a weight locally for all $n$ experts using $\Omega(n)$ memory. We stress *this is not possible in the streaming model*.

Table 3: Summary of our communication lower bounds. We assume $R \in [O(\sqrt{\frac{\log n}{T}}), O(\sqrt{\frac{n \log n}{T}})]$. All lower bounds hold against oblivious adversarial cost streams with a memory bound $M = O(\frac{n}{sTR^2} + 1)$ on the servers.

|  | LOWER BOUNDS W/ A CONSTANT PROBABILITY |
| --- | --- |
| AGG FUNC | $\ell_p (1 \leq p \leq \infty)$ |
| BROADCAST MESSAGE-PASSING | $\Omega(\frac{n}{R^2} + Ts)$ |

With $s$ servers in the message-passing model and with sum aggregation, a straightforward implementation of EWA achieves an optimal regret $O(\sqrt{\frac{\log n}{T}})$ with a trivial communication cost of $\tilde{O}(nTs)$. A distributed Exp3 algorithm achieves $O(\sqrt{\frac{n \log n}{T}})$ regret with a total communication cost of $\tilde{O}(Ts)$. Here $\tilde{O}(f)$ denotes $f \cdot \log^{O(1)}(nTs)$. A natural question is whether these bounds are tight and what the optimal regret versus communication tradeoff is.

We summarize our results in Table 1, Table 2 and Table 3. We assume $R \in [\tilde{O}((\frac{\log n}{T})^{\frac{\varepsilon}{1+\varepsilon}}), \tilde{O}((\frac{n \log n}{T})^{\frac{\varepsilon}{1+\varepsilon}})]$ for DEWA-L as well as DEWA-L-P when $1 + \varepsilon < p \leq 2$, and $R \in [\tilde{O}(\sqrt{\frac{\log n}{T}}), \tilde{O}(\sqrt{\frac{n \log n}{T}})]$ for the others. All upper bounds hold unconditionally against strong adversarial cost streams. Our upper bounds hold unconditionally against strong adaptive adversarial cost streams, where an adversary chooses its (distributed) cost vector after seeing the distribution that the algorithm uses to sample experts on that day. Also, with a memory bound on the local servers, our lower bounds hold against weaker oblivious adversarial cost streams, where the loss vectors of all days are fixed in advance. A memory-bound on individual devices, excluding the coordinator, is natural, as one should view the coordinator as a more powerful machine than the individual servers. Empirically, we also provide comprehensive evaluations over real world (HPO-B Arango et al. (2021)) as well as synthetic data traces to demonstrate the effectiveness of our methods.

## 2  Related Work

**Online learning with expert advice.** The Multiplicative Weights Update (MWU) method's first appearance dates back to the early 1950s in the context of game theory Brown & Von Neumann (1950); Brown (1951); Robinson (1951). The exact form of MWU is carried out by adding randomness, which efficiently solves two-player zero-sum games (Grigoriadis & Khachiyan, 1995). Ordentlich & Cover (1998) further proves the optimality of such algorithms under various scenarios. The algorithm has later been adopted in a wide range of applications (Cesa-Bianchi & Lugosi, 2006; Freund & Schapire, 1997; Christiano et al., 2011; Garber & Hazan, 2016; Klivans & Meka, 2017; Hopkins et al., 2020; Ahmadian et al., 2022), including the experts problem. See the comprehensive survey on MWU by Arora et al. (2012).

**Multi-armed bandits.** Similar to the experts problem, Multi-armed bandits (MAB) is another fundamental formulation in sequential optimization since its appearance in Thompson 1933; Robbins 1952. Unlike the experts problem, where each expert's cost is revealed each day, MAB limits players to observing only the cost of one expert (arm) each day. Both stochastic and adversarial MAB problems have been studied extensively (Audibert et al., 2009; Garivier & Cappé, 2011; Korda et al., 2013; Degenne & Perchet, 2016; Agrawal & Goyal, 2017; Kaufmann, 2018; Lattimore & Szepesvári, 2020; Auer et al., 2002; Auer, 2002). As we mainly consider adversarial cost streams, the Exponential-weight algorithm for Exploration and Exploitation (Exp3) and its Upper Confidence Bound (UCB) variant are most relevant due to their effectiveness in achieving near-optimal regret in the presence of adversaries (Auer et al., 2002).

**Distributed learning with expert advice.** Kanade et al. 2012 also study the expert problem under a coordinator-server model. However, the results are incomparable as Kanade et al. 2012 only considers the special case where the cost is allocated to one server rather than an arbitrary number of servers,

which makes their setup a special case under our more general scheme. Also, our lower-bound proof is against oblivious adversaries rather than adaptive adversaries, as in Kanade et al. (2012), which is more challenging to prove. Detailed comparisons with Kanade et al. (2012) are described in Section C.

Hillel et al. 2013; Szorenyi et al. 2013 give a distributed MAB setting where arms on each server share the same cost distribution, and the goal is to find the best arm cooperatively. Shahrampour et al. 2017; Landgren et al. 2016; Bistritz & Leshem 2018, on the other hand, assume the costs on each server are i.i.d. across days while being different for different servers. Cesa-Bianchi et al. 2016 considers a setup where servers are nodes on a connected graph and can only talk to neighboring nodes while restricting the cost for each arm on the servers to be the same within one day. Korda et al. 2016 studies the multi-agent linear bandit problem in a peer-to-peer network where agents share the same group of arms with i.i.d. costs across days. Some works also consider the setup where servers need to compete against each other, which is outside of our scope (Anandkumar et al., 2011; Besson & Kaufmann, 2018; Bubeck et al., 2020; Wang et al., 2020). Unlike most of these setups, we make no assumptions about the costs across days and servers.

**Distributed functional monitoring.** The coordinator-server communication model is also commonly seen in the distributed functional monitoring literature (Cormode et al., 2011; Woodruff & Zhang, 2012; Arackaparambil et al., 2009; Cormode et al., 2012; Chan et al., 2012), where the goal is to approximate function values, e.g., frequency moments, across streams with minimal communication. We note that the goal of the distributed experts problem is different in that the focus is on expert selection rather than value estimation, and the algorithms in the distributed functional monitoring literature, to the best of our knowledge, are not directly useful here.

# 3 Preliminaries and Notation

We use $T$ to denote the total number of days, $n$ the number of experts, and $s$ the number of servers. $l_{i,j}^t$ represents the cost observed at step $t$ for expert $i$ on the $j$-th server. $\hat{l}$ denotes an estimate to $l$ and $[n]$ denotes $\{1, 2, \ldots, n\}$. A word of memory is represented as $O(\log(nT))$ bits and we use $\tilde{O}(\cdot)$ to suppress $\log^{O(1)}(nTs)$ factors. We refer to the Exponential Weight Algorithm (EWA) and Multiplicative Weights Update (MWU) method interchangeably.

## 3.1 Distributed Experts Problem

In the single server expert problem, each expert $e_i, i \in [n]$ has its cost $l_i^t \in [0, 1]$ on day $t$. Based on the history, an algorithm $\mathcal{A}$ needs to select one expert $e_{\mathcal{A}(t)}$ for each day before the outcome is revealed on that day. The goal for the single server expert problem is to minimize the average regret defined as: $R(\mathcal{A}) = \frac{1}{T}\left(\sum_{t=1}^{T} l_{\mathcal{A}(t)}^t - \min_{i^*}\sum_{t=1}^{T} l_{i^*}^t\right)$.

In the distributed setting, we have $s$ servers and one coordinator where the cost $l_i^t$ now depends on costs $l_{i,j}^t$ observed locally across all the servers. The coordinator selects the expert for the next day based on any algorithm $\mathcal{A}$ of its choice. For each $j \in [s]$, the $j$-th server can receive or compute its cost $l_{i,j}^t, i \in [n]$ for the $i$-th expert on day $t$. The actual cost for the $i$-th expert on day $t$ is defined as $l_i^t = f(l_{i,1}^t, l_{i,2}^t, \cdots, l_{i,s}^t)$, where $f(\cdot)$ is an aggregation function. We assume the costs $l_{i,j}^t$ are non-negative. We consider three natural choices of $f(\cdot)$: 1. the summation function $l_i^t = \sum_{j=1}^{s} l_{i,j}^t$ and an integer power of the sum function $l_i^t = \left(\sum_{j=1}^{s} l_{i,j}^t\right)^q$ 2. the maximum/minimum function $l_i^t = \max_{j \in [s]} l_{i,j}^t$ 3. the $\ell_{p>1}$ norm function, $l_i^t = \left(\sum_{j=1}^{s}\left(l_{i,j}^t\right)^p\right)^{\frac{1}{p}}, p > 1$. In the distributed setting, regret is defined as in the single server setup with $l_i^t = f(l_{i,1}^t, l_{i,2}^t, \cdots, l_{i,s}^t)$. Without loss of generality, we normalize $l_i^t \in [0, 1], l_{i,j}^t \geq 0$. In practice, if $l_i^t \in [0, \rho]$, the regret will increase by a factor of $\rho$ accordingly, which only affects the scale of the regret and preserves optimality. Note that the cost vector for all the experts is observed by the corresponding local server. Furthermore, we explore the distributed experts problem in two different communication models:

**Message-passing model.** For the message-passing model, the coordinator can initiate a two-way private channel with a specific server to exchange messages. Messages can only be seen by the

coordinator and the selected server. The coordinator then decides which server to speak to next and repeats based on the protocol.

**Broadcast model.** In the broadcast model, the coordinator communicates with all servers using a broadcast channel. Again, the communication channel can only be initiated by the coordinator.

We further assume local servers have a memory bound of $M$ in what they can store from previous days, which is a more practical scenario as discussed in Srinivas et al. (2022); Peng & Zhang (2022). We leave the definition and description of strong adaptive adversaries and the EWA algorithm in Definition A.1 and Appendix A.2 accordingly.

# 4 Proposed Algorithms

## 4.1 Overview

In the message-passing model, we let $b_e \in [n]$ be a hyper-parameter of our choice. We first propose a baseline algorithm DEWA-S that can achieve $\tilde{O}(\sqrt{\frac{n}{Tb_e}})$ regret with constant probability using $O(T(b_e + s))$ total communication when the aggregation function is the summation function or an integer power of sum function. The intuition for the baseline algorithm is to get an unbiased estimation of the experts' underlying cost by sending a signal to the coordinator with a probability that is proportional to the local cost, which is simple yet effective. We further introduce the full algorithm DEWA-S-P that achieves $\tilde{O}(\sqrt{\frac{n}{Tb_e}})$ regret with probability $1 - \frac{1}{\text{poly}(T)}$ using $\tilde{O}(T(b_e + s))$ total communication. Both DEWA-S and DEWA-S-P work in the broadcast model with the same guarantees since the message-passing model is only more costly.

In the broadcast model, we propose DEWA-M-P that achieves $\tilde{O}(\sqrt{\frac{n}{Tb_e}})$ regret with probability $1 - \frac{1}{\text{poly}(T)}$ and using only $\tilde{O}(T(b_e + s))$ overall communication when the aggregation function is the maximum function. Besides the summation aggregation function, we leverage a random-walk-based communication protocol to find out the aggregated cost with a minimal communication cost. Since all of our protocols use (and require) at least $Ts$ communication, the coordinator can figure out the exact cost for the selected expert on each day by querying each of the $s$ servers for that expert's cost on that day. Lastly, we propose DEWA-L-P that achieves $O((\frac{n\log n}{Tb_e})^{\frac{\varepsilon}{1+\varepsilon}} + \sqrt{\frac{\log T}{T}})$ regret with probability $1 - \frac{1}{\text{poly}(T)}$ and using only $\tilde{O}(T(b_e + s))$ overall communication when the aggregation function is the $\ell_p$-norm function for any fixed constant $0 < \varepsilon \leq 1$ such that $1 + \varepsilon < p$. The algorithm employs the idea of embedding $\ell_p$ into $\ell_\infty$, thus efficiently estimating the aggregated cost using the previously introduced DEWA-M-P . For all our bounds, $b_e \in [n]$ is a hyperparameter that trades off the communication with the optimal regret we can get. For instance, setting $b_e = o(1)$ can achieve a regret of $R = \tilde{O}(\sqrt{\frac{n\log n}{T}})$ and setting $b_e = o(n)$ can achieve a regret of $R = \tilde{O}(\sqrt{\frac{\log n}{T}})$. Thus, setting $b_e = o(\frac{n}{TR^2})$ can achieve the optimal communication bound we provide in Table 1 and Table 2.

## 4.2 DEWA-S

We describe DEWA-S in Algorithm 1. The intuition is to obtain an unbiased estimate $\hat{l}^t$ for $l^t$ using limited communication and then run EWA based on our estimate. More precisely, we use the following estimator to estimate $l^t$ on day $t$: $\hat{l}_i^t = \frac{n}{b_e}(\sum_{j=1}^{s} \alpha_{i,j}^t \beta_{i,j}^t)$, where $\alpha_{i,j}^t$ are i.i.d. Bernoulli random variables following $\alpha_{i,j}^t \sim \text{Bernoulli}(\frac{b_e}{n})$, and the $\beta_{i,j}^t$ are sampled from $\text{Bernoulli}(l_{i,j}^t)$. As $l_i^t \in [0,1], l_{i,j}^t \geq 0$, $\text{Bernoulli}(l_{i,j}^t)$ is a valid distribution. We can easily verify that this is an unbiased estimator: $\mathbb{E}[\hat{l}_i^t] = \mathbb{E}[\frac{n}{b_e}(\sum_{j=1}^{s} \alpha_{i,j}^t \beta_{i,j}^t)] = \frac{n}{b_e}(\sum_{j=1}^{s} \mathbb{E}[\alpha_{i,j}^t]\mathbb{E}[\beta_{i,j}^t]) = \frac{n}{b_e}\sum_{j=1}^{s} \frac{b_e l_{i,j}^t}{n} = l_i^t$. The same sampling technique can be used to obtain an unbiased estimator of $l_i^t$ when the aggregation function is an integer power of the sum over local costs, where each monomial in the expansion of the aggregation function is unbiasedly estimated by taking the product of sampled local costs. On each day, we only incur communication cost $O(s + \sum_{i=1}^{n} \frac{b_e}{n} \sum_{j=1}^{t} l_{i,j}^t) \in O(b_e + s)$. Thus, the overall communication cost is $O(T(b_e + s))$.

---
**Algorithm 1** DEWA-S
---
**Input:** learning rate $\eta$, sampling budget $b_e$;
Initialize $\hat{L}_i^0 = 0, \forall i \in [n]$;
**for** $t = 1$ **to** $T$ **do**
    Coordinator chooses expert $i$ with probability $p(i) \propto \exp\left(-\eta \hat{L}_i^{t-1}\right)$;
    **for** $j = 1$ **to** $s$ **do**
        Coordinator initiates private channel with server $j$;
        **for** $i = 1$ **to** $n$ **do**
            Server $j$ observes cost $l_{i,j}^t$ and samples $\alpha_{i,j}^t \sim \text{Bernoulli}(\frac{b_e}{n})$, $\beta_{i,j}^t \sim \text{Bernoulli}(l_{i,j}^t)$;
            Server $j$ sends tuples $(i, j)$ to the coordinator if $\alpha_{i,j}^t = 1, \beta_{i,j}^t = 1$ and clears its memory;
    Coordinator calculates $\hat{l}_i^t = \frac{n}{b_e}(\sum_{j=1}^s \alpha_{i,j}^t \beta_{i,j}^t)$;
    Update $\hat{L}_i$ by $\hat{L}_i^t = \hat{L}_i^{t-1} + \hat{l}_i^t, \forall i \in [n]$;
---

## 4.3 DEWA-S-P

As we are using unbiased estimators instead of actual costs, we only obtain the desired regret with constant probability. In order to achieve near-optimal regret with high probability, we propose DEWA-S-P in Algorithm 2. The idea is to run multiple baseline algorithms in parallel to boost the success probability, where we regard each baseline algorithm as a meta-expert. As each meta-expert has constant success probability, the probability that they all fail is exponentially small in the number of meta-experts. Thus, by running EWA on the meta-experts, we can follow the advice of the best meta-expert and achieve near-optimal regret with high probability.

---
**Algorithm 2** DEWA-S-P
---
**Input:** learning rate $\eta_{\text{meta}}$, sampling budget $b_e$, failure rate $1/\text{poly}(T)$;
Let $K = \lceil \log\left(\text{poly}(T)\right) \rceil$, initialize $K$ baseline algorithms $\mathcal{A}_k$ and let $L_k^0 = 0, k \in [K]$;
**for** $t = 1$ **to** $T$ **do**
    Coordinator chooses expert according to $\mathcal{A}_k(t)$ with probability $p(k) \propto \exp\left(-\eta_{\text{meta}} L_k^{t-1}\right)$;
    Coordinator updates memory states for all $\mathcal{A}_k$ according to Algorithm 1;
    Coordinator receives cost $l_{\mathcal{A}_k(t)}^t = \sum_{j=1}^s l_{\mathcal{A}_k(t),j}^t$;
    Update all $L_k$ by $L_k^t = L_k^{t-1} + l_{\mathcal{A}_k(t)}^t$;
---

More precisely, to obtain $1 - \frac{1}{\text{poly}(T)}$ success probability, we initiate $\lceil \log\left(\text{poly}(T)\right) \rceil$ meta-experts $\mathcal{A}_k, k \in [\lceil \log\left(\text{poly}(T)\right) \rceil]$ at the start of the algorithm. Each meta-expert runs its own DEWA-S independently across $T$ days. The cost of the $k$-th meta-expert on day $t$ is defined to be the cost the expert $\mathcal{A}_k$ selects on the same day, which is denoted as $l_{\mathcal{A}_k(t)}^t$. With the definition of the cost for the meta-experts, we can then run EWA on the meta-experts.

The meta-level EWA needs to know the actual cost $l_{\mathcal{A}_k(t)}^t$ from the $s$ servers of each meta-expert in order to recover the best meta-expert with $1 - \frac{1}{\text{poly}(T)}$ success probability. Therefore for DEWA-S-P , on each day, we incur a communication cost of $\tilde{O}(s + (b_e + s)\log\left(\text{poly}(T)\right)) = \tilde{O}(b_e + s)$, and the overall communication is $\tilde{O}(T(b_e + s))$.

## 4.4 DEWA-M-P

We propose DEWA-M described in Algorithm 3 that achieves a near-optimal regret versus communication tradeoff up to log factors for the maximum aggregation function in the broadcast model.

The intuition of DEWA-M is that for each expert, if we walk through the servers in a random order and only update $\hat{l}_i^t$ if we encounter $l_{i,j}^t > \hat{l}_i^t$, then with high probability, we only need a small number of updates per expert. This cannot be achieved in the message-passing model due to the fact that broadcasting $\hat{l}_i^t$ requires $\Omega(s)$ communication per expert. In contrast, no communication is required for broadcasting $\hat{l}_i^t$ in the broadcast model. In fact, with probability $1 - \delta$, each expert will update at most $O(\log(s/\delta))$ times. By setting $\delta = \frac{1}{b_e \text{poly}(T)}$ and applying a union bound over our sampling budget $b_e$ and number $T$ of days, we have the desired low communication with probability at least $1 - \frac{1}{\text{poly}(T)}$. More precisely, we have the following theorem (see detailed proof in Section B.1):

---

**Algorithm 3** DEWA-M

---

**Input:** learning rate $\eta$, sampling budget $b_e$;
Coordinator initializes $\hat{L}_i^0 = 0, \forall i \in [n]$;
**for** $t = 1$ **to** $T$ **do**
  Coordinator chooses expert $i$ with probability $p(i) \propto \exp\left(-\eta \hat{L}_i^{t-1}\right)$;
  Coordinator randomly chooses $b_e$ experts with corresponding IDs $\mathcal{B}_e = \{t(1), t(2), \cdots, t(b_e)\}$;
  Coordinator initializes $\hat{l}_i^t = 0, \forall i \in [n]$;
  Coordinator permutes $[s]$ randomly and denotes the resulting sequence as $S_t$
  **for** $j$ **in** $S_t$ **do**
    Coordinator initiates channel with server $j$;
    **for** $i = 1$ **to** $n$ **do**
      Server $j$ observes cost $l_{i,j}^t$ and sends $l_{i,j}^t$ to the coordinator if $l_{i,j}^t > \hat{l}_i^t$ and $i \in \mathcal{B}_e$;
      Server $j$ cleans memory buffer;
    Coordinator updates $\hat{l}_i^t$ with received $l_{i,j}^t$;
  Update $\hat{L}_i$ by $\hat{L}_i^t = \hat{L}_i^{t-1} + \hat{l}_i^t, \forall i \in [n]$;

---

**Theorem 4.1.** *For a sampling budget $b_e \in [n]$, with probability $1 - \frac{1}{poly(T)}$, the communication cost for DEWA-M is $\tilde{O}(T(b_e + s))$.*

Even though we have a high probability guarantee with minimal communication, we still only have a constant probability guarantee for achieving optimal regret $O(\sqrt{\frac{n \log n}{b_e T}})$. We can boost the success probability using the same trick as in Algorithm 2 by initiating $\log(poly(T))$ copies of DEWA-M as meta-experts and running EWA on top of them. We refer to the high-probability version as DEWA-M-P. We thus have the following theorem (see detailed proof in Section B.2):

**Theorem 4.2.** *For a sampling budget $b_e \in [n]$, with probability $1 - \frac{1}{poly(T)}$, the communication cost for DEWA-M-P is $\tilde{O}(T(b_e + s))$.*

## 4.5 DEWA-L-P

In this section, we present DEWA-L (Algorithm 4) for the $\ell_{p>1}$ norm aggregation function in the broadcast model. The key idea of DEWA-L is to embed $\ell_p$ into $\ell_\infty$ using the min-stable property of exponential distribution. More specifically, if $E_i$ is a standard exponential random variable, then $\max_j \frac{(l_{i,j}^t)^p}{E_j} \sim \frac{(l_i^t)^p}{E}$ where $E$ is also a standard exponential random variable. Therefore, we can employ DEWA-M to efficiently compute $\frac{(l_i^t)^p}{E}$, and obtain an unbiased estimator of $l_i^t$ by normalizing.

---

**Algorithm 4** DEWA-L

---

**Input:** learning rate $\eta$, sampling budget $b_e$;
Coordinator initializes $\hat{L}_i^0 = 0, \forall i \in [n]$;
**for** $t = 1$ **to** $T$ **do**
  Coordinator chooses expert $i$ with probability $p(i) \propto \exp\left(-\eta \hat{L}_i^{t-1}\right)$;
  Coordinator randomly chooses $b_e$ experts with corresponding IDs $\mathcal{B}_e = \{t(1), t(2), \cdots, t(b_e)\}$;
  Coordinator initializes $\hat{l}_i^t = 0, \forall i \in [n]$;
  Coordinator permutes $[s]$ randomly and denotes the resulting sequence as $S_t$
  **for** $j$ **in** $S_t$ **do**
    Coordinator initiates channel with server $j$;
    Server $j$ samples $E_j \sim$ Exponential(1);
    **for** $i = 1$ **to** $n$ **do**
      Server $j$ observes cost $l_{i,j}^t$ and computes $c_{i,j}^t = \frac{(l_{i,j}^t)^p}{E_j}$;
      Server $j$ sends $c_{i,j}^t$ to the coordinator if $c_{i,j}^t > c_i^t$ and $i \in \mathcal{B}_e$;
      Server $j$ cleans memory buffer;
    Coordinator updates $c_i^t = \max_j c_{i,j}^t$ with received $c_{i,j}^t$;
  Coordinator computes $\hat{l}_i^t = \frac{1}{1-\left(1-\frac{1}{n}\right)^{b_e}} \frac{(c_i^t)^{1/p}}{\mathbb{E}\left[(E)^{-1/p}\right]}$, where $E \sim$ Exponential(1);
  Update $\hat{L}_i$ by $\hat{L}_i^t = \hat{L}_i^{t-1} + \hat{l}_i^t, \forall i \in [n]$;

---

It is not hard to see that the communication cost of DEWA-L stays the same as DEWA-M . In terms of regret, if we fix any constant $0 < \varepsilon \le 1$ such that $1 + \varepsilon < p$, DEWA-L achieves a vanishing regret $R = O((\frac{n \log n}{T b_e})^{\frac{\varepsilon}{1+\varepsilon}})$ with constant probability. Note that, for all $\ell_p$-norm functions with $p > 2$, by choosing $\varepsilon = 1$, we obtain a near-optimal regret versus communication tradeoff up to a $\log$ factor $R = O(\sqrt{\frac{n \log n}{T b_e}})$. Again, to get the high probability regret guarantee of DEWA-L , we propose DEWA-L-P that initiates $\log(\text{poly}(T))$ copies of DEWA-L as meta-experts and runs EWA on top of them. More precisely, we have the following theorem with the same proof as Theorem 4.2:

**Theorem 4.3.** *For a sampling budget $b_e \in [n]$, with probability $1 - \frac{1}{poly(T)}$, the communication cost for DEWA-L-P is $\tilde{O}(T(b_e + s))$.*

## 5 Formal Guarantees

We present formal regret analyses of DEWA-S , DEWA-S-P , DEWA-M-P and DEWA-L-P . We show that DEWA-S can achieve regret $R = O(\sqrt{\frac{n \log n}{T b_e}})$ with probability at least $9/10$, DEWA-S-P and DEWA-M-P can achieve regret $R = O(\sqrt{\frac{n \log(nT)}{T b_e}})$ with probability at least $1 - \frac{1}{\text{poly}(T)}$, and lastly DEWA-L-P can achieve regret $R = O((\frac{n \log n}{T b_e})^{\frac{\varepsilon}{1+\varepsilon}} + \sqrt{\frac{\log T}{T}})$ with probability at least $1 - \frac{1}{\text{poly}(T)}$ for any fixed constant $0 < \varepsilon \le 1$ such that $1 + \varepsilon < p$.

We then give a communication lower bound, which holds even in the broadcast model, for both summation and maximum aggregation functions with a memory bound on the individual servers. It holds for oblivious adversarial cost streams, and thus also for strong adversarial cost streams and the message-passing model. We use the communication lower bound for the $\epsilon$-DIFFDIST problem Srinivas et al. (2022) but adapt it to our setting. By reducing the $\epsilon$-DIFFDIST problem to the distributed experts problem, we prove that any protocol for achieving $R$ regret with constant probability requires total communication at least $\Omega(\frac{n}{R^2})$. It will follow that DEWA-S , DEWA-M and DEWA-L $(p > 2)$ are near-optimal in their communication for all regret values $R \in [O(\sqrt{\frac{\log n}{T}}), O(\sqrt{\frac{n \log n}{T}})]$.

### 5.1 Upper Bound

We state our regret upper bounds for DEWA-S in Theorem 5.1, DEWA-S-P in Theorem 5.2, DEWA-M-P in Theorem 5.3 and DEWA-L-P in Theorem 5.4. The detailed corresponding proofs can be found in Section B.

**Theorem 5.1.** *For $b_e \in [n]$, DEWA-S achieves regret $R = O(\sqrt{\frac{n \log n}{T b_e}})$ with probability at least $\frac{9}{10}$ for the distributed experts problem in the message passing model with the summation aggregation function and for strong adaptive adversarial cost streams.*

**Theorem 5.2.** *DEWA-S-P achieves regret $R = O(\sqrt{\frac{n \log(nT)}{T b_e}})$ with probability at least $1 - \frac{1}{poly(T)}$ for the distributed experts problem in the message passing model with the summation aggregation function and for strong adaptive adversarial cost streams.*

Notice that the total communication cost for DEWA-S-P is $\tilde{O}(T(b_e + s))$. Thus DEWA-S-P can achieve the same regret as EWA with a high probability guarantee when $b_e = n$, but requires only $\tilde{O}(T(n + s))$ communication instead of $\tilde{O}(nTs)$ communication. DEWA-S-P further generalizes to the case when $b_e < n$.

**Theorem 5.3.** *DEWA-M-P achieves regret $R = O(\sqrt{\frac{n \log(nT)}{T b_e}})$ with probability at least $1 - \frac{1}{poly(T)}$ for the distributed experts problem in the broadcast model with maximum aggregation function and for strong adaptive adversarial cost streams.*

**Theorem 5.4.** *Fix any constant $0 < \varepsilon \le 1$ such that $1 + \varepsilon < p$, DEWA-L-P achieves regret $R = O((\frac{n \log n}{T b_e})^{\frac{\varepsilon}{1+\varepsilon}} + \sqrt{\frac{\log T}{T}})$ with probability at least $1 - \frac{1}{poly(T)}$ for the distributed experts problem in the broadcast model with $\ell_p$ norm aggregation function and for strong adaptive adversarial cost streams.*

Table 4: Communication costs on the real-world HPO-B benchmark in different settings. We use EWA as the comparison baseline. E.g., DEWA-S only costs about $0.07\times$ communication of EWA.

| ALGORITHMS | EWA | EXP3 | DEWA-S | DEWA-M |
|---|---|---|---|---|
| AGG FUNC | SUM / MAX | SUM / MAX | SUM | MAX |
| SAMPLING BATCH $b_e$ | $n$ | 1 | 1 / $n$ | 1 / $n$ |
| BLACKBOARD | $1\times$ | $0.1453\times$ | $0.0730\times$ / $0.0758\times$ | $0.0849\times$ / $0.1834\times$ |
| MESSAGE-PASSING | | | | - |

Since the regret and communication bounds hold with probability at least $1 - \frac{1}{\text{poly}(T)}$ individually, by a union bound, they both hold with probability at least $1 - \frac{1}{\text{poly}(T)}$.

## 5.2 Lower Bound

**Theorem 5.5.** *Let $p < \frac{1}{2}$ be a fixed constant that is independent of the other input parameters, and suppose $M = O(\frac{n}{sTR^2} + 1)$ is an upper bound on the total memory a server can store from previous days. Any algorithm $\mathcal{A}$ that solves the distributed experts problem in the broadcast model with the $\ell_p (1 \leq p \leq \infty)$ norm aggregation function with regret $R$ and with probability at least $1 - p$, needs at least $\Omega(\frac{n}{R^2})$ bits of communication. If the algorithm can also determine, with probability at least $1 - p$, if the cost of the selected expert on each day is non-zero, then it also needs $\Omega(Ts)$ bits of communication. These lower bounds hold even for oblivious adversarial cost streams.*

We present the proof of Theorem 5.5 in Section B.7. Additionally, we present an $\Omega(ns)$ communication lower bound proof below for achieving sub-constant regret with the maximum aggregation function in the message-passing model, which is optimal for $T \in O(\text{poly}(\log(ns)))$. This indicates that we cannot do better than naïve EWA in this case, which achieves optimal regret with communication $\tilde{O}(ns)$. Note that within the optimal regret $R \in [O(\sqrt{\frac{\log n}{T}}), O(\sqrt{\frac{n \log n}{T}})]$ in which we are interested, the $T$ term can be canceled out by the $\frac{1}{T}$ term in $R^2$. So, the memory-bound assumption does not depend on the time step $T$.

## 6 Experiments

In this section, we demonstrate the effectiveness of our algorithms on the HPO-B benchmark (Arango et al., 2021) under two setups: 1. Message-passing model with summation aggregation function and 2. Broadcast model with maximum aggregation function. As a black-box hyperparameter optimization benchmark, we can regard different models in the HPO-B benchmark as different experts in the distributed experts problem, and different datasets are distributed across different servers. We further regard each search step, which is random search for all model classes, as one day in our distributed experts problem. The cost vector is then the normalized negative accuracy of models on different datasets for a search step. Thus, minimizing regret directly corresponds to optimizing the overall accuracy across all search steps. For both DEWA-S and DEWA-M , we set $b_e = 1$ to compare against Exp3 and $b_e = n$ to compare against EWA.

The results in Figure 1, Figure 2 and Table 4 show that our algorithms achieve similar regret as the optimal algorithms (Exp3 and EWA) while having less communication cost. We further use two synthetic datasets to evaluate our algorithms under various scenarios, including dense-cost and sparse-cost. We present the results in Section D, which show that our algorithms can achieve near-optimal regret with significantly lower communication cost across all scenarios consistently.

## Acknowledgements

David P. Woodruff was supported in part by a Simons Investigator Award and NSF CCF-2335412.

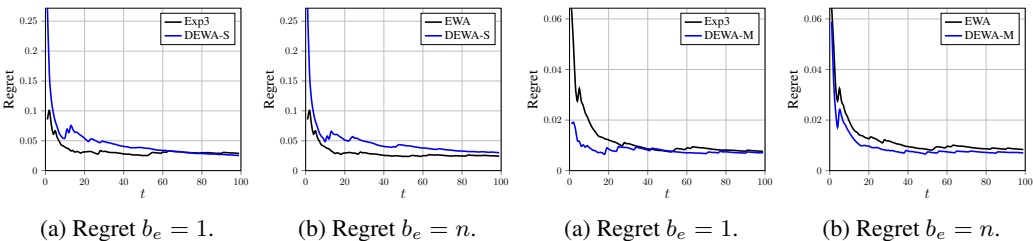

(a) Regret $b_e = 1$.      (b) Regret $b_e = n$.      (a) Regret $b_e = 1$.      (b) Regret $b_e = n$.

Figure 1: Regrets on HPO-B w/ sum aggregation. Figure 2: Regrets on HPO-B w/ max aggregation.

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

# A  Preliminaries

## A.1  Strong Adaptive Adversaries

**Definition A.1.** (Distributed experts problem with a strong adversary). An algorithm $\mathcal{A}$ run by the coordinator makes predictions for $T$ days. On day $t$:

1. $\mathcal{A}$ commits to a distribution $p_t$ over $n$ experts based on the memory contents of the coordinator on day $t$.
2. The adversary selects the cost $l_{i,j}^t$ on each server after observing $p_t$.
3. $\mathcal{A}$ selects an expert according to $p_t$ and incurs the corresponding cost.
4. The coordinator updates its memory contents by communicating with servers according to the protocol defined by $\mathcal{A}$.

We refer to adversaries that can arbitrarily define the $l_{i,j}^t$ with no knowledge of the internal randomness or state of $\mathcal{A}$, as oblivious adversaries. Notice that if we send each of the server's local information to the coordinator each day, then running the Exponential Weight Algorithm on the coordinator gives an optimal $O(\sqrt{\frac{\log n}{T}})$ regret for strong adversarial streams. However, the communication cost is a prohibitive $\tilde{O}(nTs)$ words.

## A.2  Exponential Weights Algorithm

As we will use the Exponential Weights Algorithm (EWA) as a sub-routine, we briefly describe it in Algorithm 5.   We have the following regret bound for EWA:

---
**Algorithm 5** Exponential Weight Algorithm (EWA)

---
**Input:** learning rate $\eta$;
Initialize $L_i^0 = 0, \forall i \in [n]$;
**for** $t = 1$ **to** $T$ **do**
    Sample expert $i$ with probability $p(i) \propto \exp\left(-\eta L_i^{t-1}\right)$;
    Update $L_i$ by $L_i^t = L_i^{t-1} + l_i^t, \forall i \in [n]$;

---

**Lemma A.2.** *(EWA regret, Arora et al. (2012)). Suppose $n, T, \eta > 0$, $t \in [T]$, and $l^t \in [0,1]^n$. Let $p_t$ be the distribution committed to by EWA on day $t$. Then: $\frac{1}{T}(\sum_{t=1}^{T} \langle p_t, l^t \rangle - \min_{i^* \in [n]} \sum_{t=1}^{T} l_{i^*}^t) \leq \frac{\log n}{\eta T} + \eta$. And with probability at least $1 - \delta$, the average regret is bounded by: $R(\mathcal{A}) \leq \frac{\log n}{\eta T} + \eta + O(\sqrt{\frac{\log (n/\delta)}{T}})$. Thus, taking $\eta = \sqrt{\frac{\log n}{T}}$ and $\delta = \frac{1}{poly(T)}$ gives us $O(\sqrt{\frac{\log (nT)}{T}})$ regret with probability at least $1 - \frac{1}{poly(T)}$.*

# B  Proofs.

## B.1  Theorem 4.1

In order to prove the communication bounds, we need the following lemma:

**Lemma B.1.** *Welzl (2000).  With a randomly permuted sequence $S = \{a_1, a_2, \cdots, a_n\}$ and $\gamma = 0$, if we read from left to right and update $\gamma = a_i$ whenever we encounter $a_i > \gamma$, define random variable $X$ as the number of times $\gamma$ has is updated during the process. We have the following results:*

$$\mathbb{E}\left[2^X\right] = n + 1$$

Given Lemma B.1, we can then prove our statement.

*Proof.* For any expert on any day, we will first prove that with probability at least $1 - \delta$, the servers only need to send the corresponding cost to the coordinator at most $O(\log (s/\delta))$ times. By

Lemma B.1 with $n = s$ in our setup, for any $g \geq 0$, we have:

$$
\begin{aligned}
\Pr\left(X > g\right) &= \Pr\left(2^X > 2^g\right) \\
&\leq \frac{\mathbb{E}\left[2^X\right]}{2^g} \\
&= \frac{s+1}{2^g}
\end{aligned}
$$

By setting $g = \log\left(\frac{s+1}{\delta}\right)$, we have $\Pr\left(X < \log\left(\frac{s+1}{\delta}\right)\right) > 1 - \delta$. Furthermore, letting $\delta = \frac{1}{b_e \text{poly}(T)}$, we have

$$
\Pr\left(X < \log\left((s+1)b_e\text{poly}(T)\right)\right) > 1 - \frac{1}{b_e\text{poly}(T)}.
$$

By a union bound over the $b_e$ sampled experts and $T$ days, the above guarantee simultaneously holds for all experts sampled and all days, with probability at least $1 - 1/\text{poly}(T)$. The overall communication is then:

$$
\begin{aligned}
&\sum_{t=1}^{T}\left(s + \sum_{j=1}^{b_e} X\right) \\
&\leq \sum_{t=1}^{T}\left(s + \sum_{j=1}^{b_e} \log\left((s+1)b_e\text{poly}(T)\right)\right) \\
&= Ts + Tb_e \log\left((s+1)b_e\text{poly}(T)\right) \\
&= \tilde{O}(T(b_e + s))
\end{aligned}
$$

which completes the proof. $\qquad\square$

In addition, in cases where the coordinator does not need to initiate communication, we can achieve an $O(b_e \log(s/\delta))$ communication cost per time step with the following protocol: Initialization: each individual server initializes a $\hat{h}_i^t$ to record the maximum cost for each expert. 1. For each server who has a cost larger than the current maximum, send its value to the broadcast channel after a $\delta_{i,j}$ time delay, where $\delta_{i,j}$ is randomly sampled from $[0, 1]$. 2. Once the broadcast channel has been occupied, all other servers stop the sending action and update their corresponding $\hat{h}_i^t, \delta_{i,j}$ instead. Then we can repeat this process and use the maximum value collected after $s$ unit time steps as an estimate to the maximum value. In this protocol, we assume that the broadcast channel can only be occupied by one server. The random ordering is guaranteed by the random delay and the expected number of communication rounds to get the maximum value is given in Lemma B.1. Additionally, notice that for each time step the protocol is guaranteed to end within $s$ time steps as the worst case delay is 1 unit time step for each server. By using this protocol, we can still obtain a near optimal communication cost of $O(b_e \log s/\delta)$.

## B.2  Theorem 4.2

*Proof.* Let $\mathcal{C}$ be the communication required to obtain the cost of one expert on a single day. From the proof of Theorem 4.1, we have $\mathcal{C} = O(\log\left((s+1)b_e\text{poly}(T)\right))$ with probability $1 - \frac{1}{b_e\text{poly}(T)}$. For DEWA-M-P , we need this communication bound to hold for $Tb_e \log\left(\text{poly}(T)\right) + T\log\left(\text{poly}(T)\right)$ experts and meta-experts simultaneously across a horizon of $T$ days. By a union bound, the failure rate is

$$
\frac{Tb_e \log\left(\text{poly}(T)\right) + T\log\left(\text{poly}(T)\right)}{b_e\text{poly}(T)} = 1/\text{poly}(T).
$$

As the communication cost of each expert and meta-expert is

$$
O(\log\left((s+1)b_e\text{poly}(T)\right)) = \tilde{O}(1)
$$

the overall communication cost is thus

$$
\tilde{O}(Ts + Tb_e \log\left(\text{poly}(T)\right) + T\log\left(\text{poly}(T)\right)) = \tilde{O}(T(b_e + s))
$$

with probability at least $1 - 1/\text{poly}(T)$, which concludes the proof. $\qquad\square$

## B.3 Theorem 5.1

We need the following lemmas:

**Lemma B.2.** *Define $\hat{L}_i^t = \sum_{t'=1}^t \hat{l}_i^{t'}, \hat{w}_i^t = \frac{\exp\left(-\eta \hat{L}_i^{t-1}\right)}{\sum_{i'} \exp\left(-\eta \hat{L}_{i'}^{t-1}\right)}$. Define $\hat{w}_t = [\hat{w}_1^t, \cdots, \hat{w}_n^t]^\top, \hat{l}_t = [\hat{l}_1^t, \cdots, \hat{l}_n^t]^\top$ and $\eta$ is of our choice. For all $1 \geq \varepsilon > 0$, we have the following result:*

$$\sum_{t=1}^T \langle \hat{w}_t, \hat{l}_t \rangle - \min_{i^*} \hat{L}_{i^*}^T \leq \frac{\log n}{\eta} + \frac{\eta^\varepsilon}{\varepsilon(\varepsilon+1)} \sum_{t=1}^T \sum_{i=1}^n \hat{w}_i^t (\hat{l}_i^t)^{1+\varepsilon}$$

*Proof.* Define $\Phi_t = \frac{1}{\eta} \log \left( \sum_{i=1}^n \exp\left(-\eta \hat{L}_i^t\right) \right)$

We have:

$$\Phi_T - \Phi_0$$
$$= \sum_{t=1}^T \Phi_t - \Phi_{t-1}$$
$$= \sum_{t=1}^T \frac{1}{\eta} \log \left( \frac{\sum_{i=1}^n \exp\left(-\eta \hat{L}_i^{t-1}\right) \exp\left(-\eta \hat{l}_i^t\right)}{\sum_{i=1}^n \exp\left(-\eta \hat{L}_i^{t-1}\right)} \right)$$
$$= \sum_{t=1}^T \frac{1}{\eta} \log \left( \sum_{i=1}^n \hat{w}_i^t \exp\left(-\eta \hat{l}_i^t\right) \right)$$
$$\leq \sum_{t=1}^T \frac{1}{\eta} \log \left( \sum_{i=1}^n \hat{w}_i^t \left[ 1 - \eta \hat{l}_i^t + \frac{1}{\varepsilon(\varepsilon+1)} \eta^{1+\varepsilon} (\hat{l}_i^t)^{1+\varepsilon} \right] \right)$$
$$\leq \sum_{t=1}^T \frac{1}{\eta} \sum_{i=1}^n \left( -\eta \hat{w}_i^t \hat{l}_i^t + \frac{1}{\varepsilon(\varepsilon+1)} \eta^{1+\varepsilon} \hat{w}_i^t (\hat{l}_i^t)^{1+\varepsilon} \right)$$
$$\leq -\sum_{t=1}^T \langle \hat{w}_t, \hat{l}_t \rangle + \frac{\eta^\varepsilon}{\varepsilon(\varepsilon+1)} \sum_{t=1}^T \sum_{i=1}^n \hat{w}_i^t (\hat{l}_i^t)^{1+\varepsilon}$$

where we used $\forall x \geq 0, e^{-x} \leq 1 - x + \frac{1}{\varepsilon(\varepsilon+1)} x^{1+\varepsilon}$ for the first inequality and $\forall x, \log(1+x) \leq x$ for the second inequality.

As $\Phi_0 = \frac{\log n}{\eta}$ by definition, we have:

$$\frac{\log n}{\eta} + \frac{\eta^\varepsilon}{\varepsilon(\varepsilon+1)} \sum_{t=1}^T \sum_{i=1}^n \hat{w}_i^t (\hat{l}_i^t)^{1+\varepsilon} \geq \Phi_T + \sum_{t=1}^T \langle \hat{w}_t, \hat{l}_t \rangle$$
$$\geq \sum_{t=1}^T \langle \hat{w}_t, \hat{l}_t \rangle - \min_{i^*} \hat{L}_{i^*}^T$$

where the second inequality holds due to the fact that $\forall i^* \in [n]$:

$$\Phi_T = \frac{1}{\eta} \log \left( \sum_{i=1}^n \exp\left(-\eta \hat{L}_i^t\right) \right)$$
$$\geq \frac{1}{\eta} \log \left( \exp\left(-\eta \hat{L}_{i^*}^t\right) \right)$$
$$\geq -\hat{L}_{i^*}^t$$

$\square$

**Lemma B.3.** *Let $R$ be the average regret over $T$ days. Then,*

$$\mathbb{E}[R] \leq \frac{1}{T} \cdot \mathbb{E}\left[\sum_{t=1}^{T}\langle \hat{w}_t, \hat{l}_t\rangle - \min_{i^*} \hat{L}_{i^*}^T\right]$$

*Proof.* We have:

$$
\begin{aligned}
T \cdot \mathbb{E}[R] &= \mathbb{E}\left[\sum_{t=1}^{T}\langle \hat{w}_t, l_t\rangle - \min_{i^*} L_{i^*}^T\right] \\
&= \mathbb{E}\left[\sum_{t=1}^{T}\langle \hat{w}_t, l_t\rangle\right] - \min_{i^*} L_{i^*}^T \\
&= \mathbb{E}\left[\sum_{t=1}^{T}\langle \hat{w}_t, \hat{l}_t\rangle\right] - \min_{i^*} \mathbb{E}\left[\hat{L}_{i^*}^T\right] \\
&\leq \mathbb{E}\left[\sum_{t=1}^{T}\langle \hat{w}_t, \hat{l}_t\rangle\right] - \mathbb{E}\left[\min_{i^*} \hat{L}_{i^*}^T\right] \\
&= \mathbb{E}\left[\sum_{t=1}^{T}\langle \hat{w}_t, \hat{l}_t\rangle - \min_{i^*} \hat{L}_{i^*}^T\right]
\end{aligned}
$$

Line 3 is due to $\hat{l}_t$ being independent of $\hat{w}_t$ on day $t$ and $\hat{l}_t$ is an unbiased estimator. Line 4 is by Jensen's inequality. $\qquad\square$

*Proof.* Back to our proof for Theorem 5.1, we have:

$$
\begin{aligned}
T \cdot \mathbb{E}[R] &\leq \mathbb{E}\left[\sum_{t=1}^{T}\langle \hat{w}_t, \hat{l}_t\rangle - \min_{i^*} \hat{L}_{i^*}^T\right] \quad \text{(by Lemma B.3)} \\
&\leq \mathbb{E}\left[\frac{\log n}{\eta} + \frac{\eta}{2}\sum_{t=1}^{T}\sum_{i=1}^{n}\hat{w}_i^t(\hat{l}_i^t)^2\right] \quad \text{(by Lemma B.2 with } \varepsilon = 1) \\
&= \frac{\log n}{\eta} + \frac{\eta}{2}\sum_{t=1}^{T}\mathbb{E}\left[\sum_{i=1}^{n}\hat{w}_i^t(\hat{l}_i^t)^2\right] \\
&= \frac{\log n}{\eta} + \frac{\eta}{2}\sum_{t=1}^{T}\mathbb{E}\left[\sum_{i=1}^{n}\hat{w}_i^t\mathbb{E}[(\hat{l}_i^t)^2]\right] \\
&\leq \frac{\log n}{\eta} + \frac{\eta}{2}\sum_{t=1}^{T}\mathbb{E}\left[\sum_{i=1}^{n}\hat{w}_i^t\left(\frac{2n}{b_e}\right)\right] \\
&= \frac{\log n}{\eta} + \eta\frac{Tn}{b_e}
\end{aligned}
$$

Line 5 is by:

$$\mathbb{E}\left[\left(\frac{n}{b_e}\sum_{j=1}^{s}\alpha_{i,j}^t\beta_{i,j}^t\right)^2\right]$$

$$=\frac{n^2}{b_e^2}\left(\sum_{j=1}^{s}\mathbb{E}\left[(\alpha_{i,j}^t\beta_{i,j}^t)^2\right]+\sum_{j\neq h}\mathbb{E}[\alpha_{i,h}^t\alpha_{i,k}^t\beta_{i,h}^t\beta_{i,k}^t]\right)$$

$$=\frac{n^2}{b_e^2}\left(\frac{b_e}{n}\sum_{j=1}^{s}l_{i,j}^t+\frac{b_e^2}{n^2}\sum_{j\neq h}l_{i,j}^t l_{i,h}^t\right)$$

$$\leq\frac{n^2}{b_e^2}\left(\frac{b_e}{n}+\frac{b_e^2}{n^2}(\sum_{j=1}^{s}l_{i,j}^t)^2\right)\leq\frac{2n}{b_e}$$

Take $\eta=\sqrt{\frac{b_e\log n}{Tn}}$. We then have:

$$\mathbb{E}[R]\leq 2\sqrt{\frac{n\log n}{Tb_e}}$$

Due to $R>0$, by Markov's inequality we have:

$$\Pr\left(R>20\sqrt{\frac{n\log n}{Tb_e}}\right)\leq\frac{\mathbb{E}[R]}{20\sqrt{\frac{n\log n}{Tb_e}}}\leq\frac{1}{10}$$

which concludes our proof. $\qquad\square$

### B.4 Theorem 5.2

*Proof.* Let $b_e\in[n]$ and $K=\lceil\log(\text{poly}(T))\rceil$. Let $\{\mathcal{A}_1,\mathcal{A}_2,\cdots,\mathcal{A}_K\}$ be $K$ independent DEWA-S meta-experts initiated with $b_e,b_s$. Let $\mathcal{A}_k=S$ be the event that $\mathcal{A}_k$ successfully achieves regret $O(\sqrt{\frac{n\log n}{Tb_e}})$ and let $\mathcal{A}_k=F$ be the event that it fails. From Theorem 5.1 we have:

$$\Pr(\mathcal{A}_k=F)\leq\frac{1}{10}$$

Thus, the probability that the best meta-expert achieves regret $O(\sqrt{\frac{n\log n}{Tb_e}})$ can be lower bounded by:

$$\Pr\left(\bigcup_{k=1}^{K}(\mathcal{A}_k=S)\right)\geq 1-(\frac{1}{10})^K\geq 1-1/\text{poly}(T)$$

By Lemma A.2, running EWA on top of these meta-experts gives us regret:

$$R=O(\sqrt{\frac{n\log n}{Tb_e}})+O(\sqrt{\frac{\log(K/\delta)}{T}})$$

with probability $1-1/\text{poly}(T)-\delta$ (by a union bound). Letting $\delta=1/\text{poly}(T)$ then guarantees an $O(\sqrt{\frac{n\log(nT)}{Tb_e}})$ regret with probability at least $1-\frac{2}{\text{poly}(T)}$, which concludes the proof. $\qquad\square$

### B.5 Theorem 5.3

*Proof.* For DEWA-M we have a constant probability guarantee to have regret $R=O(\sqrt{\frac{n\log(n)}{Tb_e}})$. The proof simply follows from the proof of Theorem 5.1, except that we now have actual cost for the

sampled experts instead of unbiased estimates. More specifically, we have:

$$
\begin{aligned}
T \cdot \mathbb{E}[R] &\leq \mathbb{E}\left[\sum_{t=1}^{T} \langle \hat{w}_t, \hat{l}_t \rangle - \min_{i^*} \hat{L}_{i^*}^{T}\right] \\
&\leq \mathbb{E}\left[\frac{\log n}{\eta} + \eta \sum_{t=1}^{T} \sum_{i=1}^{n} \hat{w}_i^t (\hat{l}_i^t)^2\right] \\
&= \frac{\log n}{\eta} + \eta \sum_{t=1}^{T} \mathbb{E}\left[\sum_{i=1}^{n} \hat{w}_i^t (\hat{l}_i^t)^2\right] \\
&= \frac{\log n}{\eta} + \eta \sum_{t=1}^{T} \mathbb{E}\left[\sum_{i=1}^{n} \hat{w}_i^t \mathbb{E}[(\hat{l}_i^t)^2]\right] \\
&\leq \frac{\log n}{\eta} + \eta \sum_{t=1}^{T} \mathbb{E}\left[\sum_{i=1}^{n} \hat{w}_i^t \left(\frac{n}{b_e}\right)\right] \\
&= \frac{\log n}{\eta} + \eta \frac{Tn}{b_e}
\end{aligned}
$$

Take $\eta = \sqrt{\frac{b_e \log n}{Tn}}$. We then have:

$$
\mathbb{E}[R] \leq 2\sqrt{\frac{n \log n}{Tb_e}}
$$

Since $R > 0$, by Markov's inequality we have:

$$
\Pr\left(R > 20\sqrt{\frac{n \log n}{Tb_e}}\right) \leq \frac{\mathbb{E}[R]}{20\sqrt{\frac{n \log n}{Tb_e}}} \leq \frac{1}{10}
$$

Thus, with probability at least $\frac{9}{10}$ DEWA-M has regret $R = O(\sqrt{\frac{n \log(n)}{Tb_e}})$. Since we have initiated $\log(\text{poly}(T))$ independent instances of DEWA-M , we have probability at least $1 - 1/\text{poly}(T)$ that one of the instances of DEWA-M achieves regret $R = O(\sqrt{\frac{n \log(n)}{Tb_e}})$. By Lemma A.2, running EWA on top of these meta-experts gives us regret:

$$
R = O(\sqrt{\frac{n \log n}{Tb_e}}) + O(\sqrt{\frac{\log(\log(\text{poly}(T))/\delta)}{T}})
$$

with probability $1 - 1/\text{poly}(T) - \delta$ (by a union bound). Let $\delta = 1/\text{poly}(T)$. This guarantees an $O(\sqrt{\frac{n \log(nT)}{Tb_e}})$ regret with probability at least $1 - \frac{2}{\text{poly}(T)}$, which concludes the proof. $\square$

## B.6 Theorem 5.4

*Proof.* We first upper bound the expected average regret of DEWA-L . Since $p > 1$, for any fixed constant $\varepsilon > 0$ such that $1 + \varepsilon < p$, by Lemma B.2 and Lemma B.3, we have:

$$
\begin{aligned}
T \cdot \mathbb{E}[R] & \leq \mathbb{E}\left[\sum_{t=1}^{T}\langle \hat{w}_t, \hat{l}_t \rangle - \min_{i^*} \hat{L}_{i^*}^T\right] \\
& \leq \mathbb{E}\left[\frac{\log n}{\eta} + \frac{\eta^\varepsilon}{\varepsilon(\varepsilon+1)}\sum_{t=1}^{T}\sum_{i=1}^{n}\hat{w}_i^t(\hat{l}_i^t)^{1+\varepsilon}\right] \\
& = \frac{\log n}{\eta} + \frac{\eta^\varepsilon}{\varepsilon(\varepsilon+1)}\sum_{t=1}^{T}\mathbb{E}\left[\sum_{i=1}^{n}\hat{w}_i^t(\hat{l}_i^t)^{1+\varepsilon}\right] \\
& = \frac{\log n}{\eta} + \frac{\eta^\varepsilon}{\varepsilon(\varepsilon+1)}\sum_{t=1}^{T}\mathbb{E}\left[\sum_{i=1}^{n}\hat{w}_i^t\mathbb{E}[(\hat{l}_i^t)^{1+\varepsilon}]\right] \\
& \leq \frac{\log n}{\eta} + \frac{\eta^\varepsilon}{\varepsilon(\varepsilon+1)}\sum_{t=1}^{T}\mathbb{E}\left[\sum_{i=1}^{n}\hat{w}_i^t O\left(\left(\frac{n}{b_e}\right)^\varepsilon\right)\right] \\
& = \frac{\log n}{\eta} + T \cdot O\left(\left(\frac{\eta n}{b_e}\right)^\varepsilon\right)
\end{aligned}
$$

Let $q = 1 - \left(1 - \frac{1}{n}\right)^{b_e}$ be the probability that an expert gets picked into $\mathcal{B}_e$. Line 4 is by:

$$
\begin{aligned}
\mathbb{E}\left[\left(\hat{l}_i^t\right)^{1+\varepsilon}\right] & = q \cdot \frac{1}{q^{1+\varepsilon}} \cdot \frac{\mathbb{E}\left[(c_i^t)^{(1+\varepsilon)/p}\right]}{\mathbb{E}^{1+\varepsilon}\left[E^{-1/p}\right]} \\
& = q^{-\varepsilon}\frac{\mathbb{E}\left[(l_i^t)^{1+\varepsilon} \cdot E^{-(1+\varepsilon)/p}\right]}{\mathbb{E}\left[E^{-1/p}\right]} \\
& = q^{-\varepsilon} \cdot \left(l_i^t\right)^{1+\varepsilon} \cdot \frac{\mathbb{E}\left[E^{-(1+\varepsilon)/p}\right]}{\mathbb{E}\left[E^{-1/p}\right]} \\
& \leq q^{-\varepsilon} \cdot \frac{\mathbb{E}\left[E^{-(1+\varepsilon)/p}\right]}{\mathbb{E}\left[E^{-1/p}\right]} \quad (\text{as } 0 \leq l_i^t \leq 1) \\
& = O\left(q^{-\varepsilon}\right) \quad (\text{as } \mathbb{E}\left[E^{-(1+\varepsilon)/p}\right] \text{ and } \mathbb{E}\left[E^{-1/p}\right] \text{ converge}) \\
& = O\left(\left(\frac{n}{b_e}\right)^\varepsilon\right)
\end{aligned}
$$

Pick $\eta = \left(\frac{b_e}{n}\right)^\varepsilon \cdot \frac{\log n}{\varepsilon T}$, we then have:

$$
T \cdot \mathbb{E}[R] = O\left(T^{\frac{1}{1+\varepsilon}}\left(\frac{n\log n}{b_e}\right)^{\frac{\varepsilon}{1+\varepsilon}}\right)
$$

Hence,

$$
\mathbb{E}[R] = O\left(\left(\frac{n\log n}{Tb_e}\right)^{\frac{\varepsilon}{1+\varepsilon}}\right)
$$

By Markov's inequality, DEWA-L has an average regret $R = O\left(\left(\frac{n\log n}{Tb_e}\right)^{\frac{\varepsilon}{1+\varepsilon}}\right)$ with probability at least $\frac{9}{10}$.

Since we have initiated $\log(\text{poly}(T))$ independent instances of DEWA-L , we have probability at least $1 - 1/\text{poly}(T)$ that one of the instances of DEWA-L achieves regret $R = O\left(\left(\frac{n\log n}{Tb_e}\right)^{\frac{\varepsilon}{1+\varepsilon}}\right)$.

By Lemma A.2, running EWA on top of these meta-experts gives us regret:

$$R = O\left(\left(\frac{n \log n}{Tb_e}\right)^{\frac{\varepsilon}{1+\varepsilon}}\right) + O\left(\sqrt{\frac{\log\left(\log\left(\text{poly}(T)\right)/\delta\right)}{T}}\right)$$

with probability $1 - 1/\text{poly}(T) - \delta$ (by a union bound). Let $\delta = 1/\text{poly}(T)$. This guarantees an $O\left(\left(\frac{n \log n}{Tb_e}\right)^{\frac{\varepsilon}{1+\varepsilon}} + \sqrt{\frac{\log T}{T}}\right)$ regret with probability at least $1 - \frac{2}{\text{poly}(T)}$, which concludes the proof.

$\square$

## B.7 Lower Bound Proof

The communication lower bound proof for the maximum aggregation function in the message-passing model follows using the multi-player number-in-hand communication lower bound for set disjointness in Braverman et al. (2013). To solve the multi-player set disjointness problem with $s$ players, where each player has $n$ bits of information $c_i^j \in \{0, 1\}, i \in [n], j \in [s]$, the communication lower bound is $\Omega(ns)$ for the message-passing model.

In our problem, in the first case, all experts have at least one server that has a cost of 1, i.e., $\exists j \in [s], \forall i \in [n], c_i^j = 1$. In the second case, we have one expert whose cost on every server is 0 while the other experts all have at least one server that has a cost of 1. Then, in the first case, the sets (cost vectors on each server) are disjoint for all coordinates (experts) while in the second case, there exists one coordinate (expert) whose intersection over all sets is non-empty. In the second case, this expert has a maximum cost of 0 while all other experts incur a maximum cost of 1. If we can decide which case we are in, then we solve the set disjointness problem, and thus there is an $\Omega(ns)$ communication bound. By copying the same hard instance over $T$ days, it follows that if there exists an algorithm that can achieve sub-constant regret for this distributed experts problem, then the algorithm also solves the above set disjointness problem. We have thus obtained an $\Omega(ns)$ communication bound for the maximum aggregation function in the message-passing model. Note that EWA can achieve the optimal regret with $\tilde{O}(ns)$ communication if we assume $T \in O(\text{poly}(\log(ns)))$, and therefore, we cannot do better than EWA up to logarithmic factors with the maximum aggregation function in the message-passing model. To give the lower bound proof, we first define the $\epsilon$-DIFFDIST problem.

**Definition B.4.** ($\epsilon$-DIFFDIST problem, Srinivas et al. (2022)). There are $T$ players, and each has $n$ bits of information indexed from 1 to $n$. Let $\mu_0 = \text{Bernoulli}(\frac{1}{2})$, $\mu_1 = \text{Bernoulli}(\frac{1}{2} - \epsilon)$, we must distinguish between the following two cases:

- (Case A). Each index for each player is drawn i.i.d. from $\mu_0$.
- (Case B). An index $i \in [n]$ is randomly chosen, then the $i$-th indexed bit of each player is drawn i.i.d. from $\mu_1$ while other bits of players are all drawn i.i.d. from $\mu_0$.

**Lemma B.5.** ($\epsilon$-DIFFDIST communication bound, Srinivas et al. (2022)). The communication complexity of solving the $\epsilon$-DIFFDIST problem with a constant $1 - p$ probability under the broadcast model, for any $p \in [0, 0.5)$, is $\Omega(\frac{n}{\epsilon^2})$

Note that a lower bound for the broadcast model is also a lower bound for the message-passing model. By regarding different days as servers and bits as cost streams of experts, if we generate bits from either case A or case B, then the algorithm needs to distinguish between case A and case B to obtain regret at most $\epsilon$. We design Algorithm 6 to connect the $\epsilon$-DIFFDIST with the distributed experts problem. Algorithm 6 gives a reduction from $\epsilon$-DIFFDIST, and thus we obtain our lower bound in Theorem 5.5. The additional $Ts$ factor is from our requirement that we obtain an approximation to the actual cost for the selected expert on each day. We present the complete proof as follows:

*Proof.* [1] We will prove this by showing for $R = \frac{1}{2 + \sqrt{2 \ln(24)}}$ and $p = \frac{1}{3}$, Algorithm 6 can indeed solve $\epsilon$-DIFFDIST with probability at least $\frac{2}{3}$. The proof extends naturally to any constant $\delta, p < \frac{1}{2}$.

---

[1]The proof follows Srinivas et al. (2022) with a different model and objective.

**Algorithm 6** An algorithm that reduces the $\epsilon$-DIFFDIST to the summation-based distributed experts problem in the broadcast model.

---

**Input:** $\{X^1, \cdots, X^t, \cdots, X^T\}$, where $X^t \in \{0,1\}^n$ for all $t \in [T]$ is a binary vector generated from $\epsilon$-DIFFDIST; Oracle algorithm $\mathcal{A}$ that solves the summation-based distributed experts problem with regret $R$ and probability larger than $\frac{1}{2}$;

**Let** $c = \sqrt{2\ln(24)}, \epsilon = R(c+1) < 1/2$;

**Cost definition:** For day $t$, we randomly sample a server $j$ and define $l_j^t = X^j$ and $l_{j'}^t = \mathbf{0}, \forall j' \in [s]/\{j\}$;

**Initialize** $M_0$ as the initial memory state on the coordinator for $\mathcal{A}$, counter $C = 0$;

**for** $t = 1$ **to** $T$ **do**
    Obtain the actual cost $l(t) = \mathcal{A}(M_{t-1})$ incurred by $\mathcal{A}$;
    $C$ += $l(t)$;
    Update memory state to $M_t$ by communicating with downstream servers according to $\mathcal{A}$;

**Let** $\hat{C} = \frac{C}{T}$ be the average cost;

**if** $\hat{C} > \frac{1-Rc}{2}$ **then**
    **Return** Case A;

**else**
    **Return** Case B;

---

We further need $R < \frac{1}{2(c+1)}$ to make sure $\frac{1}{2} + \epsilon$ is a valid probability. Let $\hat{C}$ be the average cost of $\mathcal{A}$. We will show we can solve the $\epsilon$-DIFFDIST problem in both cases.

For case A, $\hat{C}$ is just the average of $T$ i.i.d. coin flips. Thus, by Hoeffding's inequality we have:

$$
\begin{aligned}
\Pr\left(\hat{C} \leq \frac{1-Rc}{2}\right) &= \Pr\left(1 - \hat{C} \geq \frac{1+Rc}{2}\right) \\
&\leq \exp\left(-\frac{TR^2c^2}{2}\right) \\
&\leq \exp\left(-\frac{c^2}{2}\right) \\
&< \frac{1}{3}
\end{aligned}
$$

where the third line is due to $TR^2 \geq 1$.

For case B, let $C^*$ be the average cost of the expert whose cost is generated from $\mu_1 = \text{Bernoulli}(\frac{1}{2} - R(c+1))$. As we know, $\mathcal{A}$ has the guarantee that $\hat{C} \leq C^* + R$ with probability at least $\frac{3}{4}$, so we have:

$$
\begin{aligned}
\Pr(\hat{C} > \tfrac{1-Rc}{2}) & \\
\leq \Pr\left(\left(\hat{C} > C^* + R\right) \cup \left(C^* + R > \tfrac{1-Rc}{2}\right)\right) & \\
\leq \Pr\left(\hat{C} > C^* + R\right) + \Pr\left(C^* + R > \tfrac{1-Rc}{2}\right) & \\
\leq \tfrac{1}{4} + \Pr\left(C^* > \tfrac{1}{2} - R(c+1) + \tfrac{Rc}{2}\right) & \\
\leq \tfrac{1}{4} + \exp\left(-\tfrac{TR^2c^2}{2}\right) & \\
< \tfrac{1}{3} &
\end{aligned}
$$

Thus we have shown that we can solve the $\epsilon$-DIFFDIST problem in both cases with probability at least $\frac{2}{3}$, and therefore make Algorithm 6 a valid reduction. As a result, the total communication cost of Algorithm 6 is at least $\Omega(\frac{n}{R^2})$ by Lemma B.5. In addition, if we need to know the cost of the selected expert, we need to pay an extra $\Omega(s)$ communication per day. Indeed, we need $\Omega(s)$ communication even if we just want to verify whether the selected expert incurs zero cost or not with probability larger than $\frac{9}{10}$. This is due to the fact that we can choose our distribution so that on each day, we choose a random server and with probability $1/2$ make the cost $0$ on that server, while with the remaining probability $1/2$ we make the cost $1$ on that server. All other servers have cost $\mathbf{0}$. Thus, if the protocol probes $o(s)$ servers on each day, it only has a $1/2 + o(1)$ probability to know if the cost is non-zero or not. Thus, we need to at least probe $\Omega(s)$ servers to succeed with constant

probability on a single day, and since the days are independent, $\Omega(sT)$ communication in total. Thus, we overall have a communication lower bound of $\Omega(\frac{n}{R^2} + Ts)$.

Since we allow each server to have $M = O(\frac{n}{sTR^2} + 1)$ memory, we can actually save communication for messages sent between the same server but on different days. However, the communication required can be reduced by at most $TMs$. Let $\text{Cost}(A)$ be the communication cost for $\mathcal{A}$. We then have $\text{Cost}(A) + TMs \in \Omega(\frac{n}{R^2} + Ts)$. As $TMs \in O(\frac{n}{R^2} + Ts)$, we thus have $\text{Cost}(A) \in \Omega(\frac{n}{R^2} + Ts)$, which completes the proof. $\qquad\square$

For the maximum/$\ell_p$ norm aggregation function in the broadcast model, we can use the same proof with the same bound since the maximum/$\ell_p$ norm operation gives us the same cost streams as the summation operation under our setting where one random server has cost $X^t$ while others have zero costs.

## C   Comparison with Kanade et al. (2012)

Although we address a similar topic with Kanade et al. (2012), we would like to stress that our setup differs quite significantly. In our setup, the ground truth costs for experts are aggregated across all servers. In contrast, the setup of Kanade et al. (2012) restricts the ground truth costs for each expert to be allocated to exactly one server per day. Consequently, our setup is more general since instead of finding out the only server that carries the cost on each day, we also incur additional costs from other servers as well. In addition, Kanade et al. (2012) only proves their lower bound for $n = 2$ while we handle general $n$. On the other hand, for $n = 2$, they show a lower bound for adaptive adversaries rather than oblivious adversaries, which is our setting. However, we also make an assumption on the server memory budget for proving lower bounds. In fact, our lower bound directly matches that of Kanade et al. (2012) when $n = 2$ if we do not require the coordinator or current transcript to dictate who speaks next as the additive $Ts$ term is no longer needed. More specifically, we compare in Table 5 for the case when only the coordinator can initiate conversation and in Table 6 for the case when both the coordinator and servers can initiate conversation.

Table 5: Coordinator initiates message-passing channel

|  | Lower Bound | Upper Bound |
|---|---|---|
| Ours | $\Omega(n/R^2 + Ts)$ and oblivious adversaries | $\tilde{O}(n/R^2 + Ts)$ |
| Kanade et al. (2012) | $\Omega(1/R^2)$ for $n = 2$ and adaptive adversaries | Not applicable |

Table 6: Coordinator or server initiates message-passing channel

|  | Lower Bound | Upper Bound |
|---|---|---|
| Ours | $\Omega(n/R^2)$ for any $n$ and oblivious adversaries | $\tilde{O}(n/R^2)$ |
| Kanade et al. (2012) | Not applicable | Not applicable |

Note that we can remove the $Ts$ term if the servers are allowed to spontaneously initiate conversation, in which case synchronization between servers on each day is not required. We note that Kanade et al. (2012)'s upper bound is not applicable in our setting as it assumes the cost (payoff vector) to be distributed to only one server. At the same time, we allow the cost to be distributed to any number of servers. Thus, their setup is a special case of ours. We note that our bounds also match those of Kanade et al. (2012) in this special case, e.g., our upper bound is also $\tilde{O}(\frac{n}{R^2})$. In short, our results are incomparable as we allow: 1. Costs to be distributed to any number of servers 2. Any $n$ for the lower-bound proof against oblivious adversaries rather than adaptive adversaries.

Table 7: Communication costs of constant-probability protocols on Gaussian distribution in different settings. We use EWA as the comparison baseline.

| ALGORITHMS | EWA | EXP3 | DEWA-S | BASE-S | DEWA-M | BASE-M |
|---|---|---|---|---|---|---|
| AGG FUNC | SUM / MAX | SUM / MAX | SUM | SUM | MAX | MAX |
| SAMPLING BATCH $b_e$ | $n$ | 1 | 1 / $n$ | 1 / $n$ | 1 / $n$ | 1 / $n$ |
| BROADCAST (NON-SPARSE) | $1\times$ | $0.0196\times$ | $0.0099\times$ / $0.0203\times$ | $0.0104\times$ / $0.0298\times$ | $0.0104\times$ / $0.0503\times$ | $0.0145\times$ / $0.7328\times$ |
| MESSAGE-PASSING (NON-SPARSE) | | | | | - | - |
| BROADCAST (SPARSE) | $1\times$ | $0.0196\times$ | $0.0099\times$ / $0.0203\times$ | $0.0104\times$ / $0.0298\times$ | $0.0100\times$ / $0.0188\times$ | $0.0039\times$ / $0.0198\times$ |
| MESSAGE-PASSING (SPARSE) | | | | | - | - |

# D  Simulated Experiments

**Evaluation setup.** In this section, we evaluate the performance of DEWA-S and DEWA-S-P with the summation aggregation function, and DEWA-M and DEWA-M-P with the maximum aggregation function. We measure the average regrets over the days and total communication costs and compare the performance with EWA when $b_e = n$, and with Exp3 when $b_e = 1$. We further evaluate two cost distributions, namely, the Gaussian and Bernoulli distributions. On each server, the costs of the experts are randomly sampled from these distributions. For the best expert, the costs are sampled from $\mathcal{N}(0.2, 1)$ or $Bernoulli(0.25)$, and for the other experts, the costs are sampled from $\mathcal{N}(0.6, 1)$ or $Bernoulli(0.5)$. For the summation aggregation, all of the costs are truncated to the range $[0, 1]$ and then divided by the number of servers $s$. To show the robustness of our protocols under extreme cost conditions, we also evaluate a scenario where the costs are sparsely distributed across the servers, i.e., the cost of an expert is held by one server, and other servers receive zero cost for that expert. To further emphasize the effectiveness of our protocol design in such sparse scenarios, we implement and evaluate the performance of the simplified DEWA-S and DEWA-M and we treat them as BASE-S and BASE-M along with their high probability versions BASE-S-P and BASE-M-P . We describe the detail of the baseline algorithms in the following section. We set the learning rate $\eta = 0.1$, the number of servers to be $s = 50$, the number of experts to be $n = 100$, and the total days to be $T = 10^5$ for $b_e = 1$ and to be $T = 10^4$ for $b_e = n$. We set the sampling budget $b_s = 2$ for BASE-S and BASE-S-P . The experiments are run on an Ubuntu 22.04 LTS server equipped with a 12 Intel Core i7-12700K Processor and 32GB RAM.

## D.1  Baselines

For baselines to be compared, we use the simplified variants of DEWA-S and DEWA-M , namely BASE-S and BASE-M . More specifically, for BASE-S , instead of sampling according to cost values, BASE-S is set to sample servers uniformly. The estimate of cost $l_i^t$ is then defined as:

$$\hat{l}_i^t = \frac{ns}{b_e} \sum_j \alpha_{i,j}^t \beta_{i,j}^t l_{i,j}^t,$$

where $\alpha_{i,j}^t \sim \text{Bernoulli}(\frac{b_e}{n}), \beta_{i,j}^t \sim \text{Bernoulli}(\frac{1}{s})$. This is a good baseline to compare with since $\hat{l}_i^t$ is also an unbiased estimator. However, due to the uniform sampling strategy, BASE-S will fail in the sparse setting and require an additional factor of $s$ in the regret while DEWA-S does not suffer from this.

For BASE-M , we uniformly sample among servers and take the maximum cost encountered as the estimate of the actual cost $l_i^t$. To illustrate the effectiveness of DEWA-M , we enforce that the overall communication cost for BASE-M is close to DEWA-M when $b_e = 1$ or $b_e = n$.

## D.2  Results of Gaussian Distribution Cost

In Figure 3, we first present the regrets of DEWA-S and DEWA-S-P on the Gaussian distribution with the summation aggregation function in the non-sparse setting. As we can see in Figure 3a, with sampling budget $b_e = 1$, DEWA-S achieves much smaller regrets than Exp3. And the protocols' average regrets over $t$ are converging to 0 with increasing $t$. The regrets of all the protocols are comparable to that of EWA when the sampling budget $b_e = n$, as shown in Figure 3b. However, for the sparse scenario, as shown in Figure 4, the regrets of DEWA-S and DEWA-S-P are much better than BASE-S and BASE-S-P . When $b_e = 100$, DEWA-S and DEWA-S-P can still achieve comparable performance to EWA in the sparse setting. The results further illustrate that our design is

Table 8: Communication costs of high-probability protocols on Gaussian distribution in different settings. We use EWA as the comparison baseline.

| ALGORITHMS | EWA | EXP3 | DEWA-S-P | BASE-S-P | DEWA-M-P | BASE-M-P |
|---|---|---|---|---|---|---|
| AGG FUNC | SUM / MAX | SUM / MAX | SUM | SUM | MAX | MAX |
| SAMPLING BATCH $b_e$ | $n$ | 1 | 1 / $n$ | 1 / $n$ | 1 / $n$ | 1 / $n$ |
| BROADCAST (NON-SPARSE) | 1× | 0.0196× | 0.4829× / 0.5822× | 0.4945× / 0.7624× | 0.0781× / 0.1975× | 0.0729× / 0.7718× |
| MESSAGE-PASSING (NON-SPARSE) | | | | | - | - |
| BROADCAST (SPARSE) | 1× | 0.0196× | 0.4829× / 0.5823× | 0.4945× / 0.7623× | 0.0596× / 0.0862× | 0.0706× / 0.1182× |
| MESSAGE-PASSING (SPARSE) | | | | | - | - |

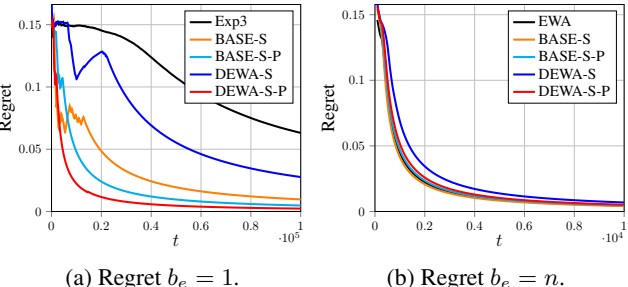

(a) Regret $b_e = 1$.  (b) Regret $b_e = n$.

Figure 3: Regrets on Gaussian distribution with summation aggregation, non-sparse scenario.

effective and can handle such extremely sparse cost conditions. As expected, the high-probability versions of the protocols consistently achieve lower regret than their constant-probability versions.

For the maximum aggregation function, we observe similar results as shown in Figure 5 and Figure 6. The regrets of DEWA-M and DEWA-M-P are close to EWA when $b_e = n$, and their performance is much better than Exp3 when $b_e = 1$. We also observe that the regrets of BASE-M and BASE-M-P are close to that of DEWA-M and DEWA-M-P in the non-sparse setting. However, their communication costs are much higher than DEWA-M and DEWA-M-P when $b_e = n$, as shown in Table 7 and Table 8. Consistent with our findings for the summation aggregation function, in the sparse setting, the regrets of BASE-M and BASE-M-P are much higher than DEWA-M and DEWA-M-P . The results illustrate that DEWA-M and DEWA-M-P are not restricted to i.i.d. costs among the servers, and they work well in extremely sparse settings. Thus, we conclude that DEWA-M and DEWA-M-P have wider application scopes.

We report our communication costs for constant a probability guarantee in Table 7 and for a high probability guarantee in Table 8. We use the communication cost of EWA as the baseline (1×), which is $\tilde{O}(nTs + Ts)$. According to our results, for $b_e = 1$ and $b_e = n$, DEWA-S and DEWA-S-P use much smaller communication than Exp3 and EWA respectively. We also notice that, in the sparse setting, DEWA-M and DEWA-M-P use much smaller communication to achieve near-optimal regret, since DEWA-M and DEWA-M-P can quickly identify the server holding large costs. Although the BASE counterparts achieve comparable communication costs to DEWA-S , DEWA-S-P , DEWA-M , and DEWA-M-P , considering their much larger regret in the sparse setting, DEWA-S , DEWA-S-P , DEWA-M , and DEWA-M-P are more consistent across settings. By increasing $b_e$, the protocols achieve lower regret at the cost of more communication. Users can choose $b_e$ according to their regret requirements and communication budget. Even if we set $b_e = n$, the communication costs are still much smaller than that of EWA, but the regret of our algorithms is very close to optimal.

### D.3  Results of Bernoulli Distribution Cost

In this section, we present our regret and communication on Bernoulli distributed costs. Our regrets are shown in Figure 7, Figure 8, Figure 9, and Figure 10 and our communication costs are presented in Table 9 and Table 10, which are consistent with our observations for Gaussian distribution. DEWA-S , DEWA-S-P , DEWA-M , and DEWA-M-P all perform well in both non-sparse and sparse scenarios, with near-optimal regrets and much smaller communication costs compared with the EWA.

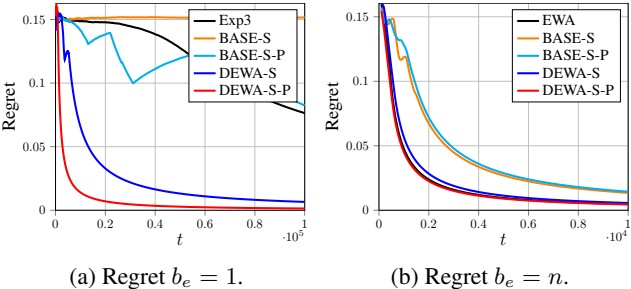

(a) Regret $b_e = 1$.  (b) Regret $b_e = n$.

Figure 4: Regrets on Gaussian distributions with summation aggregation, sparse scenario.

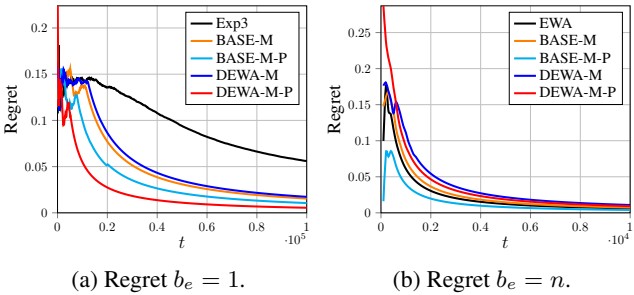

(a) Regret $b_e = 1$.  (b) Regret $b_e = n$.

Figure 5: Regret on Gaussian distribution with maximum aggregation, non-sparse scenario.

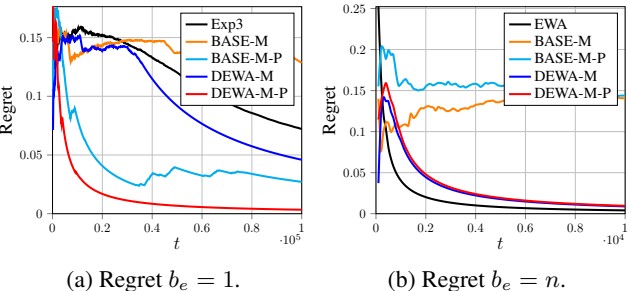

(a) Regret $b_e = 1$.  (b) Regret $b_e = n$.

Figure 6: Regret on Gaussian distribution with maximum aggregation, sparse scenario.

Table 9: Communication costs of constant-probability protocols on Bernoulli distribution in different settings. We use EWA as the comparison baseline.

| ALGORITHMS | EWA | EXP3 | DEWA-S | BASE-S | DEWA-M | BASE-M |
|---|---|---|---|---|---|---|
| AGG FUNC | SUM / MAX | SUM / MAX | SUM | SUM | MAX | MAX |
| SAMPLING BATCH $b_e$ | $n$ | 1 | 1 / $n$ | 1 / $n$ | 1 / $n$ | 1 / $n$ |
| BROADCAST (NON-SPARSE) | $1\times$ | $0.0196\times$ | $0.0099\times$ / $0.0196\times$ | $0.0104\times$ / $0.0298\times$ | $0.0102\times$ / $0.0376\times$ | $0.0145\times$ / $0.7328\times$ |
| MESSAGE-PASSING (NON-SPARSE) | | | | | - | - |
| BROADCAST (SPARSE) | $1\times$ | $0.0196\times$ | $0.0099\times$ / $0.0196\times$ | $0.0104\times$ / $0.0298\times$ | $0.0099\times$ / $0.0160\times$ | $0.0039\times$ / $0.0198\times$ |
| MESSAGE-PASSING (SPARSE) | | | | | - | - |

Table 10: Communication costs of high-probability protocols on Bernoulli distribution in different settings. We use EWA as the comparison baseline.

| ALGORITHMS | EWA | EXP3 | DEWA-S-P | BASE-S-P | DEWA-M-P | BASE-M-P |
|---|---|---|---|---|---|---|
| AGG FUNC | SUM / MAX | SUM / MAX | SUM | SUM | MAX | MAX |
| SAMPLING BATCH $b_e$ | $n$ | 1 | 1 / $n$ | 1 / $n$ | 1 / $n$ | 1 / $n$ |
| BROADCAST (NON-SPARSE) | $1\times$ | $0.0196\times$ | $0.4827\times$ / $0.5676\times$ | $0.4945\times$ / $0.7623\times$ | $0.0483\times$ / $0.1303\times$ | $0.0729\times$ / $0.7718\times$ |
| MESSAGE-PASSING (NON-SPARSE) | | | | | - | - |
| BROADCAST (SPARSE) | $1\times$ | $0.0196\times$ | $0.4827\times$ / $0.5677\times$ | $0.4945\times$ / $0.7624\times$ | $0.0397\times$ / $0.0579\times$ | $0.0706\times$ / $0.1182\times$ |
| MESSAGE-PASSING (SPARSE) | | | | | - | - |

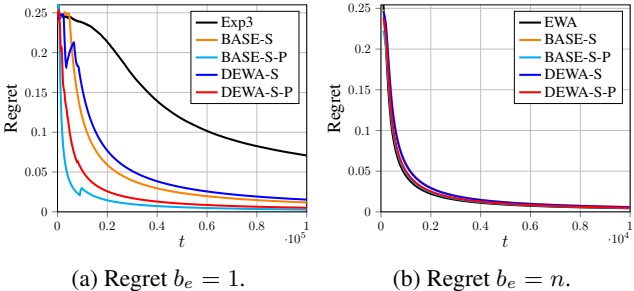

(a) Regret $b_e = 1$.      (b) Regret $b_e = n$.

Figure 7: Regrets on Bernoulli distribution with summation aggregation, non-sparse scenario.

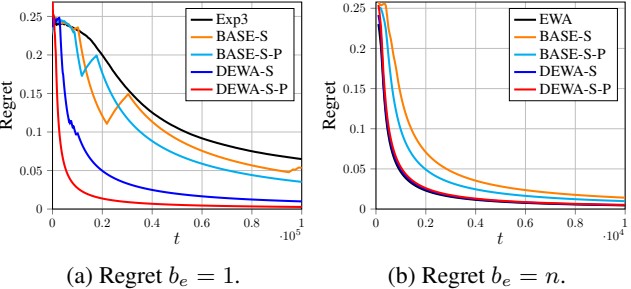

(a) Regret $b_e = 1$.      (b) Regret $b_e = n$.

Figure 8: Regrets on Bernoulli distribution with summation aggregation, sparse scenario.

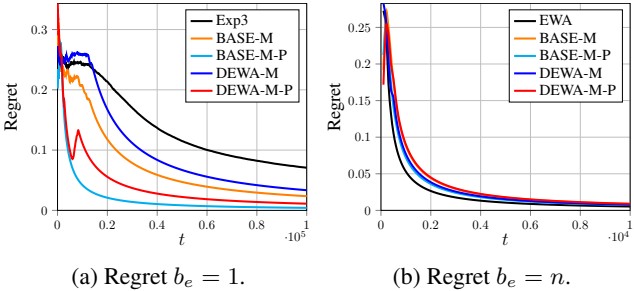

(a) Regret $b_e = 1$.      (b) Regret $b_e = n$.

Figure 9: Regret on Bernoulli distribution with maximum aggregation, non-sparse scenario.

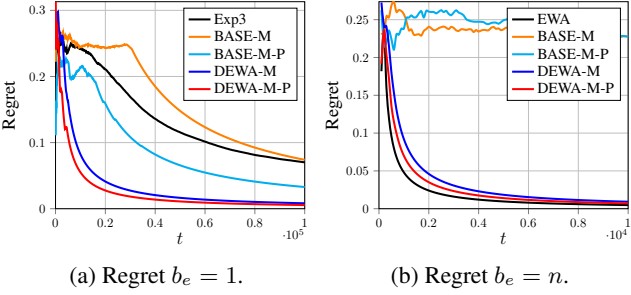

(a) Regret $b_e = 1$.      (b) Regret $b_e = n$.

Figure 10: Regret on Bernoulli distribution with maximum aggregation, sparse scenario.

### D.4 Evaluation Results under Different $b_e$

To further study the influence of $b_e$ on our algorithms, we evaluate the regret and communication cost of DEWA-S-P and DEWA-M-P under different $b_e$, ranging from 1 to $n = 100$. The results on the regret results can be found in Figure 11. As expected, using a larger $b_e$ makes the regret converge faster. We observe that using a reasonably large value ($0.25n$ in our experiments) is sufficient to achieve good regret. The resulting communication cost using different $b_e$ can be found in Figure 12. As expected, the cost generally grows linearly with respect to increasing $b_e$.

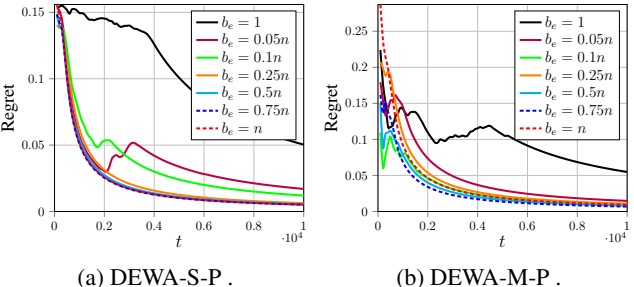

(a) DEWA-S-P .  (b) DEWA-M-P .

Figure 11: Regret for Gaussian distribution under different $b_e$, non-sparse scenario.

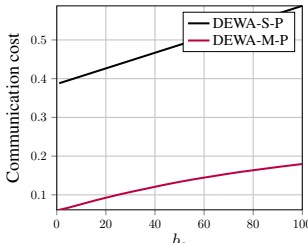

Figure 12: Communication cost of DEWA-S-P and DEWA-M-P using different $b_e$, and with EWA as the baseline, non-sparse scenario.

