# OpenReview forum: "Communication Bounds for the Distributed Experts Problem"
_NeurIPS.cc/2024/Conference — NeurIPS 2024 poster_

### Official Review · Reviewer_rqWP · 2024-07-10

**Soundness:** 2
**Presentation:** 2
**Contribution:** 1
**Rating:** 6
**Confidence:** 3

**Summary:**

The paper considers a distributed variant of the classical problem of learning with experts, where the cost of each expert needs to be aggregated across different servers. Based on three different aggregation models, i.e., sum, maximum and $\ell\_p$ norm,  the authors propose three different algorithm based on the classical exponential weights algorithm. The authors derive bounds on regret and communication costs of the three proposed algorithms. Authors also derive matching lower bounds to demonstrate optimality of their results. Theoretical results are supplemented by empirical results.

**Strengths:**

It is an interesting idea to consider the distributed version of this classical problem, especially as applications of distributed continue to grow. The use of randomization in max aggregation and $\ell\_lp$ norms (through exponential random variable) is interesting.

**Weaknesses:**

The novelty of the paper especially in terms of analysis is limited. The analysis of all the algorithms is largely based on existing results.

Please refer to the next section for additional questions.

**Questions:**

I was a reviewer of this paper during ICML 2024 where I had raised some questions which the authors did not answer. Since the content of the paper is largely similar to the ICML submission of this work, my following questions continue to remain.

1. Why does the coordinator need to initiate communication? In both the models, it is assumed that the coordinator needs to initiate the communication in each round. This is not a very typical consideration in a distributed setting. Usually, there is uplink channel that the clients can use to communicate whenever they want and downlink channel to broadcast messages to the clients/servers. While it might seem that this is a minor issue, I am not convinced that is the case. Firstly, I think it is reason as to why the communication costs have the $Ts$ term. If the clients can themselves initiate the communication, then these terms go away both in the upper and lower bounds. However, it is important to note that if such a coordination does not exist, then the randomization scheme used for aggregation via maximization does not work, or equivalently incurs a cost of $\mathcal{O}(s + b\_e\log(s/\delta))$ while the lower bound would be independent of $s$. This rather unnatural setting is somehow implicitly important for the algorithm to achieve optimal communication cost in the maximum aggregation model.

2. I think there is misrepresentation of results/implicit assumption in the sum aggregation model, which the authors should highlight. There is an implicit assumption that the loss vectors are sparse, in the sense that loss for any expert is high value only for small set of servers. In the experimental section, the authors claim that their results hold "even under extreme settings of sparse loss vectors" while the truth is their theoretical results hold _only_ for sparse regime. Note that the authors claim that in the sum aggregation model the loss for $i^{\text{th}}$ expert at time $t$ is given as $l_{i}^t = \sum_{j = 1}^{s} l_{i,j}^t$ and that the loss is normalized to ensure $l_{i}^t \in [0,1]$. This implicitly places a constraint that $\sum_{j = 1}^{s} l_{i,j}^t \leq 1$ forcing sparser loss vectors. The sparsity is crucial to obtain the improved communication guarantees. A natural choice for "normalization" (as the authors mention) is to average the total loss, i.e., $l_{i}^t = \frac{1}{s} \sum_{j = 1}^{s} l_{i,j}^t$ to ensure $l_{i}^t \in [0,1]$. This is also something the authors have done in the experiments themselves. While such an averaging step does not change the regret guarantees, it significantly impacts the communication cost as the communication cost is proportional to $b_e \cdot T \cdot \sum_{j = 1}^{s} l_{i,j}^t$. If $l_{i}^t = \frac{1}{s} \sum_{j = 1}^{s} l_{i,j}^t \in [0,1]$ then $ \sum_{j = 1}^{s} l_{i,j}^t \leq s$ implying that the worst case communication cost is $\mathcal{O}(b_e T s)$ as opposed to $\mathcal{O}(T(b_e + s))$. I think this is very important point that the authors have not elaborated upon and can change the significance of the results.

**Limitations:**

Yes.

---

> ### Author Rebuttal · Authors · 2024-08-06
>
> We thank the reviewer for their constructive comments and address the main concerns below.
>
> **Q1. The coordinator assumption**
>
> We agree with the reviewer that in some scenarios there does not exist a coordinator to coordinate the communications among downstream servers. The motivation for the coordinator initialization assumptions is: 1. to avoid the complexities in asynchronous communications, e.g., handshake protocols or extra buffers to guarantee ordering; 2. the coordinator is commonly seen in both message-passing and broadcast models in practice as well as theory [1, 2]. Additionally, for the case where the coordinator does not need to initiate communication, we can achieve an $O(b_e \log{(s/\delta)})$ communication cost per time step with the following protocol: Initialization: each individual server initializes a $\hat{h_i^t}$ to record the maximum cost for each expert. 1. For each server who has a cost larger than the current maximum, send its value to the broadcast channel after a $\delta_{i, j}$ time delay, where $\delta_{i, j}$ is randomly sampled from $[0, 1]$. 2. Once the broadcast channel has been occupied, all other servers stop the sending action and update their corresponding $\hat{h_i^t}, \delta_{i, j}$ instead. Then we can repeat this process and use the maximum value collected after $s$ unit time steps as an estimate to the maximum value. In this protocol, we assume that the broadcast channel can only be occupied by one server. The random ordering is guaranteed by the random delay and the expected number of communication rounds to get the maximum value is given in Lemma B.1. Additionally, notice that for each time step the protocol is guaranteed to end within $s$ time steps as the worst case delay is 1 unit time step for each server. By using this protocol, we can still obtain a near optimal communication cost of $O(b_e \log{s/\delta})$. We thank the reviewer for bringing this up and inspiring us to design a new protocol under different assumptions. We will add the new results and setups in the next version.
>
> **Q2. Sparsity for the summation aggregation function**
>
> First we want to point out that normalizing a cost vector to $l_i^t \in [0, 1], l_{i, j}^t \geq 0$ is a common practice in the experts problem literature. The way we distribute $l_i^t$ to different servers is purely random, which does not impose any sparsity assumption. In fact, even if $l_i^t \in [0, \rho], l_{i, j}^t \geq 0, \forall \rho > 0$, the communication upper bound for our algorithm will not change as our sampling probability (more specifically $\beta_{i, j}^t$) is designed to be $\frac{l_{i, j}^t}{\rho}$, and then the communication cost is proportional to $b_e \cdot T \cdot \frac{\sum_{j=1}^s l_{i, j}^t}{\rho}$, where $\frac{\sum_{j=1}^s l_{i, j}^t}{\rho} \in [0, 1]$. As we have assumed that $\rho=1$ in our setup without affecting the optimality, we have ignored the $\rho$ factor during our derivation, which might cause the confusion. We thank the reviewer for pointing this out and will make our definitions and explanations clear to avoid potential confusion.
>
> **References:**
>
> 1. Kanade, Varun, Zhenming Liu, and Bozidar Radunovic. "Distributed non-stochastic experts." Advances in Neural Information Processing Systems 25 (2012).
>
> 2. Braverman, Mark, and Rotem Oshman. "On information complexity in the broadcast model." Proceedings of the 2015 ACM Symposium on Principles of Distributed Computing. 2015.

---

> > ### Comment · Reviewer_rqWP · 2024-08-09
> > **Response to Authors**
> >
> > Thank you for your response.
> >
> > Response to coordinator assumption: The proposed fix seems interesting and should fix the issue. Can the authors please elaborate a bit on why do they need the additional random delay?
> >
> > Response to sparsity of aggregation function: I think I misunderstood that $\beta$ will need to be updated accordingly. That should resolve the concern.

---

> > > ### Author Response · Authors · 2024-08-10
> > > **Clarifications by Authors**
> > >
> > > We thank the reviewer for additional feedback.
> > >
> > > The random delay provides randomness to the protocol, so that in each round, there can only be a single random server who can occupy the broadcast channel. Randomness is required to guarantee this upper bound against the strongest adversary. Indeed, otherwise, the adversary can exploit the communication pattern and compromise the correctness of the protocol.  For instance, if we fix the server communication order to be from 1 to s, then the adversary can exploit the pattern by assigning $s$ monotonically increasing cost values to the $s$ downstream servers, which,  because of our fixed ordering, requires $s$ rounds of communications per expert. Thus, the overall communication cost would be $\tilde{\Omega}{(Tns)}$, which is much worse than the optimal communication cost we have if we incorporate randomness in our algorithm by using random delays.

---

> > > > ### Comment · Reviewer_rqWP · 2024-08-10
> > > > **Response to the authors**
> > > >
> > > > Ah I see, that makes sense. Thanks a lot for the clarification. I have updated my score accordingly.

---

> > > > > ### Author Response · Authors · 2024-08-11
> > > > > **Thanks for the updates**
> > > > >
> > > > > We really thank the reviewer for the effort and time in reviewing as well as bringing insightful suggestions, which we will surely incorporate in the final paper.

---

### Official Review · Reviewer_eSR8 · 2024-07-10

**Soundness:** 3
**Presentation:** 3
**Contribution:** 2
**Rating:** 5
**Confidence:** 4

**Summary:**

The paper investigates the experts' problem in a distributed context, where the costs associated with experts are distributed across multiple servers. The authors present results for two communication models: the message-passing model and the blackboard model. They explore two aggregation functions: the sum and max of an expert's cost across servers. The paper introduces communication-efficient protocols designed to achieve near-optimal regret in these settings.

The paper considers two communication models: the message-passing model, which involves two-way communication channels, and the blackboard model, which utilizes a broadcast channel. The objective is to balance near-optimal regret and communication efficiency.

The proposed algorithms leverage sampling techniques to approximate the aggregated functions and reduce communication overhead.

**Strengths:**

The paper examines an important problem, thoroughly covering upper bounds, lower bounds, and empirical evaluations.

**Weaknesses:**

The paper acknowledges the contributions of Kanade et al. in related works. However, it relegates the discussion of the connection with this work to the appendix. This is somewhat unexpected, as the mentioned work shares a significant relation to the current study, probably more so than other cited literature.

The paper considers two aggregation functions: sum and max. The selection of max, however, seems a bit odd and lacks clear motivation.

The lower bound in this paper requires a limit on the memory the central server can utilize. Such a situation is not common in communication lower bounds. This condition seems to be more characteristic of the lower bounds for streaming algorithms rather than for communication models.

**Questions:**

Could you move the discussion comparing your work with the Kanade et al. paper into the main text?

Could you provide more justification for selecting the 'max' function as one of the aggregation functions in your study?

---

> ### Author Rebuttal · Authors · 2024-08-06
>
> We thank the reviewer for their review and address their main concerns below.
>
> **Q1. Comparison with Kanade et al.**
>
> We thank the reviewer for the suggestion. We will move the comparison with Kanade et al. to the main text in a revision, as we indeed consider it very relevant to our work.
>
> **Q2. Max aggregation function**
>
> The maximum aggregation function is usually used when the objective is dependent on the worst cost across different servers, e.g., if the objective is to lower the worst case serving latency for some downstream tasks to satisfy certain SLOs (Service Level Objectives), or we want to bound the maximum drawdown for a diversified investment across different regions. Notice that besides the summation and maximum aggregation function, we also support the $l_p$ aggregation function, which is a much more general aggregation function, where the maximum function is a very special case $l_\infty$ as we assume all costs are positive. We believe the $l_p$ aggregation function protocols also introduce several interesting technical ideas.
>
> **Q3. Memory bound assumption**
>
> We thank the reviewer for pointing out our memory assumption on the lower bound. First, we note that our memory bound is only imposed on downstream servers with no memory bound assumption on the central coordinator. We also note that our lower bound is against the weakest possible adversary (oblivious adversarial) and thus holds in the most general setting. On the other hand, notice that the overall memory requirement for all downstream servers is $O(\frac{n}{TR^2}+s)$ which increases linearly as we have more servers. This does model situations in which the memory budget is constant for each server.

---

> > ### Comment · Reviewer_eSR8 · 2024-08-14
> >
> > Thank you for the response. It is satisfactory. I will maintain my score.

---

### Official Review · Reviewer_B3J6 · 2024-07-12

**Soundness:** 3
**Presentation:** 3
**Contribution:** 3
**Rating:** 7
**Confidence:** 4

**Summary:**

I reviewed this paper a few years back a few times and declined to review this paper for a while to be fair to the authors. I found the paper’s quality has not improved much (as opposed to the average quality improvement for many recycled theory papers) so I can only give somewhere between borderline and accept vote again this time. I would like to emphasize that contains interesting insights to distributed expert problems and could easily be a solid Neurips work. For example, it considers a very generic setting where the aggregation function is $\ell_p$ for all $p$, and its lower bound that utilizes communication complexity also appears to be quite interesting (as opposed to the standard expert lower bound that relies on anti-concentration). But the writing really makes it difficult to champion the paper.

**Strengths:**

It performs a set of interesting theoretical exercise to a fundamental expert problem in the distributed setting. Techniques developed in this work are useful addition to this area.

**Weaknesses:**

I feel more effort is needed to make this paper a smoother read for both theoreticians and general audience.

Some (writing) issues I found:

1. It was not even completely clear if this is a multi-arm problem or an expert problem from Sec. 3.1. It looks like an expert problem but the authors mentioned the multi-arm setting quite early on.
For the broadcasting setting, how is cost counted? Number of broadcasts?
2. Tables 1 to 3 are not really very comprehensible. R is left unexplained but it looks like a tunable parameter related to regret but Table 3 has a bound for R whereas Table 1 and 2 do not have.
3. Notation like R \in [O(something), O(something)] is not really rigorous as the authors are probably aware themselves. Maybe just be explicit to write down constants, or at least do $O(something) \cap \Omega(something)$.
4. The lower bound on the space constraint $M$ also seems not carefully thought out, or parameterized in a quite awkward manner. The space has to shrink as the number of servers grows?

Those are all quite minor stuff so I wish the authors could make some effort to clear them up.

**Questions:**

I do not have any questions.

**Limitations:**

Not applicable.

---

> ### Author Rebuttal · Authors · 2024-08-06
>
> We thank the reviewer for their review and address their main concerns below.
>
> **Q1. Presentation**
>
> We thank the reviewer for pointing out the issues in our presentation.
>
> The problem is indeed an experts problem as we assume each server can observe the full cost vector. The reason we introduced the multi-arm bandit problem in the related work is that the optimal algorithm (Exp3) proposed for the multi-arm bandit problem can serve as one possible solution for our problem and thus a baseline that we can compare with.
>
> We thank the reviewer for pointing out the inconsistency of assumptions on $R$ between the different tables. In the last paragraph of Section 1 we mention the assumptions: we assume $R \in [ \tilde{O}  ( (\frac{\log n}{T} )^{\frac{\varepsilon}{1+\varepsilon}}  ), \tilde{O}  ( (\frac{n\log n}{T} )^{\frac{\varepsilon}{1+\varepsilon}}  )  ]$ for DEWA-L as well as DEWA-L-P when $1+\varepsilon < p \leq 2$, and $R \in  [\tilde{O}(\sqrt{\frac{\log{n}}{T}}), \tilde{O}(\sqrt{\frac{n\log{n}}{T}}) ]$ for the others. We will make the regret bound clearer and more consistent across all tables.
>
> We thank the reviewer for pointing out the notational confusion. The reason why we write the regret R to be in a range is that our algorithm allows for a hyper-parameter $b_e \in [n]$, which can be specified to give a tradeoff between the regret and the communication cost. Depending on the different choice of $b_e$ our algorithm can achieve a corresponding optimal regret $R=O(\sqrt{\frac{n\log{n}}{T b_e}})$. Thus, given that $b_e \in [n]$, the optimal regret $R will be in $[O(\sqrt{\frac{\log{n}}{T}}), O(\sqrt{\frac{n\log{n}}{T}})]$ accordingly. To avoid confusion, we will make this point clearer in the preliminaries.
>
> **Q2. Memory bound**
>
> We thank the reviewer for pointing out our memory assumption on the lower bound. First, we note that our memory bound is only imposed on downstream servers. We also note that our lower bound is against the weakest possible adversary (oblivious adversarial) and thus holds in the most general setting. Here we obtain an $O(\frac{n}{sTR^2}+1)$ bound, which for  the near-optimal regrets $R \in [O(\sqrt{\frac{\log{n}}{T}}), O(\sqrt{\frac{n\log{n}}{T}})]$ we consider, we have that the $\frac{1}{\sqrt{T}}$ term cancels with the $T$ in our $TR^2$ bound. We will make the memory bound clearer in our revision to avoid unnecessary confusion.
>
> The memory constraint on each server will indeed become stronger as the number of servers increases due to more communication required to achieve the optimal regret. It is a challenging open question to completely remove this assumption. On the other hand, notice that the overall memory requirement for all servers is $O(\frac{n}{TR^2}+s)$ which increases linearly as we have more servers. This does model situations in which the memory budget is constant for each server.

---

### Official Review · Reviewer_FQkW · 2024-07-14

**Soundness:** 3
**Presentation:** 2
**Contribution:** 3
**Rating:** 6
**Confidence:** 3

**Summary:**

This paper studies the classical experts setting in a new communication-focused model that is motivated by evaluating models when the data points are stored across many different servers.  At a high level, the model is the following.  There are $n$ experts (think of them as different models).  There are $s$ servers, each of which is presumably storing some data.  At time $t$, there is some $l_{i,j}^t$ for the $i$'th expert on server $j$.  The overall loss for expert $i$ at time $t$ is $f(l_{i,1}^t, \dots, l_{i,s}^t)$ for some aggregation function $f$.  Natural choices of the aggregation function, which this paper studies, are the sum, the max, and more generally $\ell_p$-norms.  The distributed nature comes from the fact that we, the coordinator, do not have access to the server-specific losses unless we ask them and so incur a communication cost.  So we might have only partial information about each expert, unless we query all servers for all experts.  And since communication is an important bottleneck, we naturally want to minimize it.  So we have an obvious question: what is the tradeoff between communication and achievable regret?

As the authors point out, there are two obvious algorithms.  First, we could simply ask every server for the loss of every expert in every round.  This would incur $O(ns)$ communication per round, but would allow us to run any standard experts algorithm to get regret $O(\sqrt{\frac{\log n}{T}})$.  On the other hand, we could use a *bandit* algorithm like Exp3 to select a single expert at each time, and query every server for that one expert.  This would incur $O(s)$ communication in each round, but would give regret bounds of $O(\sqrt{\frac{n\log n}{T}})$, i.e., the standard bandit regret bound.  These algorithms would work for any aggregation function, since they collect full information for whatever experts they are considering.  So the natural question is whether it is possible to do better: for example, can we get the communication of the bandit algorithm ($O(s)$ per round) with the regret bound of the expert algorithm ($O(\sqrt{\frac{\log n}{T}})$)?

This paper also distinguishes between two communication models, message-passing and broadcast.  In message-passing, sending a message from the coordinator to a server (or vice versa) has a cost of 1 (or the number of words).  In broadcast, by contrast, we assume that the coordinator and the receiver are on a shared channel, so sending one word to *everyone* costs $1$.

This paper gives algorithms and some lower bounds for the natural aggregation functions in this model.  At a very high level, in message-passing they can only handle the sum aggregation function, but has regret like the experts setting and communication $O(n+s)$ per round.  So if $n \leq s$, they get the communication of the bandit setting but the regret of the expert setting, getting the best of both worlds.  They also get a similar bound for the max aggregation function, but only in the broadcast communication model.

**Strengths:**

Overall, I like this paper and would advocate acceptance, although it has a few weaknesses.  But fundamentally, it's an interesting problem and they give reasonable results.

- The motivating example of distributed online optimization seems quite reasonable to me, and more generally I like the idea of "evaluating different experts requires talking to many different servers".  While there has been previous work on "distributed experts", this paper is (to the best of my knowledge) the first to study this particular setting.

- The given algorithms are reasonably simple, which is great, but are not obvious.  The main idea is that we need to evaluate the quality of an expert without actually querying every server about them.  They do this probabilistically, getting an unbiased estimator of the cost of each expert for very cheap and then using this estimator.  This has some of my favorite type of analysis, where the math isn't "hard", but there are clever and non-obvious ideas.

**Weaknesses:**

In my opinion, there are two main weaknesses of this paper: some of the motivation and assumptions of the results, and some of the writing.

- The most interesting (and first) result is for the sum aggregation and the message-passing model, but for the other aggregation functions they need the broadcast model.  And I don't understand the motivation for this model.  They discuss how it is a standard model of a one-hop wireless network, which is totally true, but they don't talk at all about why we might have this kind of distributed expert problem in a one-hop wireless network.  And I can't think of any plausible story for this.  Instead, the obvious motivation is their first one on distributed online optimization, which presumably is happening in a datacenter and so is a natural fit for the message-passing model.  So I found it a bit disappointing that most of their results need a communication model that they do not really justify as being plausible for the problem that they're trying to solve.

- The lower bound makes very strange assumptions.  In particular, it assumes that the memory at each server is bounded by a function that depends *inversely* on both $T$ and the regret $R$.  Why would this be?  I don't understand why there would be any such dependence, particularly as a function of $T$.  If anything, one might imagine the memory *growing* with $T$ (as the stream gets longer we buy more and more space on the server); I certainly don't understand why it would be *shrinking*.

- The writing in this paper is quite poor, in a few different ways.
  - At a high level, the authors do a poor job of explaining what they're trying to do (get expert-style regret bounds with bandit-style communication bounds).  I spent a long time thinking that they were trying to get improved dependence on $T$, rather than changing the dependence on $n$.  I'll note that I usually think of $n$ (the number of experts) as being fixed and $T$ going to infinity, so I personally have never cared too much about the difference in the $n$ dependence between experts and bandits.  It makes sense for $n$ to be large in the distributed online optimization setting, though, so I actually do believe that this is an interesting result.  They just don't explain it well at all.
  - At a more detailed level related to the above point, they basically do not explain or give context to their results at all.  They discuss their results starting on line 73, but basically just say "our results are in Tables 1, 2, and 3".  They don't discuss how we should interpret these bounds at all -- are they good?  bad?  Is there room for improvement?  What's tight and what's not?  How do they relate to non-distributed classical settings?  This is just crazy writing -- I've never seen such a lack of discussion about the results in any paper.  This is one of the main reasons why I found it so hard to understand the importance of the results (see above).
  - At an even more detailed level, even the results tables are strange.  First, they don't seem to actually prove these bounds anywhere.  Look, for example, at the communication bound in Table 1 for DEWA-S.  As far as I can tell, this bound does not appear in any theorem, corollary, or lemma in the rest of the paper.  Instead, the bound for DEWA-S that they actually prove is Theorem 5.1 for the regret (they don't have a theorem statement about the communication anywhere, although they argue it in Section 4.2).  So there is a mismatch between the bounds in the intro and the bounds in the paper.  Of course, the bounds in the intro can be derived from the bounds that they actually prove, but for some reason they do not actually do this, or discuss why they phrase the bounds in these two different ways.
  - Similarly, if you look at the results table or the theorem statements, since they are hiding logarithmic terms the bounds for the high probability results are identical to the constant probability results.  So what's the point of the constant probability results?  Why not just give the high probability results as your results, and the constant probability algorithms as useful subroutines for the high probability case?

**Questions:**

- What is the argument for why the broadcast communication model is reasonable for this particular problem (not other distributed problems)?

- Why does the lower bound assume memory that decreases with $R$ and $T$?  What's the motivation for this, and why is it a plausible and interesting scenario?

**Limitations:**

This is fine.

---

> ### Author Rebuttal · Authors · 2024-08-06
>
> We thank the reviewer for their review and address their main concerns below.
>
> **Q1. Motivation for the Broadcast Model**
>
> We thank the reviewer for acknowledging the setup for the message-passing model. For the broadcast model, there are fewer scenarios for the distributed experts problem than in the message-passing model, but there are still several important ones. Imagine a multi-level distributed online learning setting, where the upper-level consists of message-passing models while the lower-level in the hierarchy is a one-hop wireless model. For example, consider an online federated learning setting in which the end servers are edge devices such as cell phones. Our algorithm can be used in different levels within the hierarchy based on different communication models to achieve optimal performance. Additionally, a satellite communication model is another scenario in which the broadcast model is important. Other than practical motivations, the broadcast model is a well-studied model [1, 2] across different domains including streaming [3, 4, 7], cryptography [5], and mechanism design [6]. We therefore introduce the broadcast model for the distributed experts problem and provide optimal communication and regret tradeoffs under various aggregation functions such as the maximum and the $l_p$ norm.
>
> **Q2. Lower Bound**
>
> We thank the reviewer for pointing out our memory assumption on the lower bound. First, we note that our memory bound is only imposed on downstream servers. We also note that our lower bound is against the weakest possible adversary (oblivious adversarial) and thus holds in the most general setting. Here we obtain an $O(\frac{n}{sTR^2}+1)$ bound, which for  the near-optimal regrets $R \in [O(\sqrt{\frac{\log{n}}{T}}), O(\sqrt{\frac{n\log{n}}{T}})]$ we consider, we have that the $\frac{1}{\sqrt{T}}$ term cancels with the $T$ in our $TR^2$ bound. We will make the memory bound clearer in our revision to avoid unnecessary confusion.
>
> **Q3. Writing**
>
> We thank the reviewer for pointing out the issues in our presentation. The communication bound for DEWA-S is derived in Section 4.2, but we will also include the formal theorem and proof of the communication bound for DEWA-S to align with DEWA-M and DEWA-L. We will also make the notation of the paper clearer and more consistent.
>
> **References:**
>
> 1. Braverman, Mark, and Rotem Oshman. "On information complexity in the broadcast model." Proceedings of the 2015 ACM Symposium on Principles of Distributed Computing. 2015.
>
> 2. Kushilevitz, Eyal. "Communication complexity." Advances in Computers. Vol. 44. Elsevier, 1997. 331-360.
>
> 3. Noga Alon, Yossi Matias, and Mario Szegedy. The space complexity of approximating the frequency moments. Journal of Computer and System Sciences, 58(1):137 – 147, 1999.
>
> 4. Ziv Bar-Yossef, T. S. Jayram, Ravi Kumar, and D. Sivakumar. An information statistics approach to data stream and communication complexity. J. Comput. Syst. Sci., 68(4):702–732, 2004.
>
> 5. Shahar Dobzinski, Noam Nisan, and Sigal Oren. Economic efficiency requires interaction. In Proceedings of the 46th Annual ACM Symposium on Theory of Computing, STOC ’14, pages 233–242, 2014.
>
> 6. Oded Goldreich and A Warning. Secure multi-party computation. unpublished manuscript, 1998.
>
> 7. André Gronemeier. Asymptotically optimal lower bounds on the NIH-multi-party information complexity of the AND-function and disjointness. In Proc. 26th Symp. on Theor. Aspects of Comp. Sc. (STACS), pages 505–516, 2009

---

> > ### Comment · Reviewer_FQkW · 2024-08-07
> >
> > I've read the rebuttal, and have the following comments and questions.
> >
> > **Q1. Motivation for Broadcast**
> >
> > I totally agree with the authors that broadcast is a standard model in theoretical computer science; I have certainly written papers in a variety of different broadcast models.  But I still don't really buy it for this particular setting.  In your example of a federated hierarchical system where only the bottom level is broadcast, it seems to me like if that's the actual setting that is motivating this, you should explicitly model it and argue that doing well in the bottom layer is sufficient (maybe because the cost incurred in upper layers is negligible)?  My intuition is that all of the settings discussed in the rebuttal (and the paper) for broadcast "feel" like the stories that theorists tell ourselves in order to focus on the math without thinking too hard about the applications, rather than actual applications or motivations.  And that is, to me, a weakness (although note I still like the paper and think it should be accepted).
> >
> > **Q2. Lower Bound**
> >
> > If I understand your rebuttal correctly, you're saying that due to your assumption on $R$ the memory upper bound is really something like $O(n / (s \log n))$ (for the smallest regret) or $O(1/(s\log n) + 1)$ (for the largest regret).  Is that what you're saying?  That makes a little more sense than my initial reaction.  But this still seems to me to be a very strong assumption on the memory (particularly in the large regret case).  Is there some justification for it?  Why should I think that the servers are limited to this much memory?

---

> > > ### Author Response · Authors · 2024-08-08
> > > **Clarifications by Authors**
> > >
> > > We thank the reviewer for additional feedback.
> > >
> > > For **Q1**, we thank the reviewer for acknowledging our theoretical contribution and agree that the broadcast model is mostly brought up in the field of theoretical computer science as a natural, though arguably less practical model. Nevertheless, we hope our derived algorithms can provide insight for settings such as distributed online learning in the satellite communication model, or for abstracting the communication costs that we may be interested in, e.g., the coordinator sends a single message to a router which lists multiple destinations, and we only count the cost incurred at the coordinator.
> > >
> > > For **Q2**, yes, the memory bound is $O(\frac{n}{s\log{n}}+1)$ for the smallest regret and $O(\frac{1}{s\log{n}}+1)$ for the largest regret, but note that the memory assumption is only required for our lower bound argument and is only needed for the downstream servers. The reason we pose such a memory bound in our lower bound proof is that the general case (without the memory bound) is a hard communication complexity problem and we currently do not have a solution for the most general case. On the other hand, our memory bound always allows for at least any constant amount of memory for each downstream server, which is practical in scenarios in which the memory budget for each server is limited. We hope our initiation of the study of this problem in this model can inspire further work to close the gap in the fully general setting.

---

### Decision · Program_Chairs · 2024-09-25

**Decision:**

Accept (poster)

**Comment:**

This paper studies the experts problem in the distributed setting, where the costs associated with experts are distributed across multiple servers.  The authors give both upper and lower bounds.

The overall sentiment is generally favorable towards acceptance. Most reviewers found the problem under study to be interesting and the results to be reasonable. The authors' response has increased the reviewers' confidence in the paper, particularly by effectively addressing one reviewer's concerns about the various assumptions made, such as a space constraint is needed for proving the lower bound.